_Article_

# Osmotic signaling releases PP2C-mediated inhibition of _Arabidopsis_ SnRK2s via the receptor-like cytoplasmic kinase BIK1

Guo-Jun Li[1,2,3,5], Kong Chen[1,2,4,5], Shujing Sun[1] & Yang Zhao [1,2,3]✉

## Abstract

**Osmotic stress and abscisic acid (ABA) signaling are important for plant growth and abiotic stress resistance. Activation of osmotic and ABA signaling downstream of the PYL-type ABA receptors requires the release of SnRK2 protein kinases from the inhibition imposed by PP2Cs. PP2Cs are core negative regulators that constantly interact with and inhibit SnRK2s, but how osmotic signaling breaks the PP2C inhibition of SnRK2s remains unclear. Here, we report that an _Arabidopsis_ receptor-like cytoplasmic kinase, BIK1, releases PP2C-mediated inhibition of SnRK2.6 via phosphorylation regulation. The dominant _abi1-1_ ABA-signaling mutation (G180D) disrupts PYL-PP2C interactions and disables PYL-initiated release of SnRK2s; in contrast, BIK1 releases _abi1-1_-mediated inhibition of SnRK2.6. BIK1 interacts with and phosphorylates SnRK2.6 at two tyrosine residues, which are critical for SnRK2.6 activation and function. Phosphorylation of the two tyrosine residues may affect the docking of the tryptophan "lock" of PP2C into SnRK2.6. Moreover, the _bik1_ mutant is defective in SnRK2 activation, stress-responsive gene expression, ABA accumulation, growth maintenance, and water loss under osmotic stress. Our findings uncover the critical role of BIK1 in releasing PP2C-mediated inhibition of SnRK2s under osmotic stress.**

**Keywords** Osmotic Stress; SnRK2 Release; BIK1; Tyrosine Phosphorylation; PP2C
**Subject Category** Plant Biology

## Introduction

To survive under drought and osmotic stress, terrestrial plants have evolved osmoregulatory mechanisms, such as activation of SNF1-related protein kinase 2s (SnRK2s) (Boudsocq et al, 2004; Fujii et al, 2011), accumulation of abscisic acid (ABA), and activation of ABA signaling (Zhu, 2016). Since osmoregulation often antagonizes plant growth and development (Zhang et al, 2020), osmotic signaling is blocked under well-watered conditions, mainly through the action of type 2C protein phosphatases (PP2Cs) from clade A (Umezawa et al, 2009). PP2Cs are the evolutionarily conserved negative regulators of osmoregulation and SnRK2s (Komatsu et al, 2013; Rubio et al, 2009; Tougane et al, 2010; Umezawa et al, 2009). PP2Cs constantly interact with SnRK2s and inhibit them by dephosphorylating the serine in their activation loop and blocking the catalytic cleft (Soon et al, 2012; Umezawa et al, 2009) (Fig. 1A, left panel). During osmotic stress or in response to ABA signaling, this inhibition by PP2Cs must be released for any activation of SnRK2s and downstream osmoregulation (Soma et al, 2023).

The two-step activation mechanism of SnRK2s has been well-characterized in the context of ABA signaling (Fig. 1A, middle panel). The first step is implemented by the Pyrabactin resistance 1 (PYR1) and PYR1-like (PYL) proteins [hereafter referred to as PYLs, also named regulatory components of the ABA receptors (RCARs)]. The PYLs serve as ABA receptors and, upon binding ABA, competitively interact with PP2Cs, which releases SnRK2s from PP2C-mediated inhibition (Ma et al, 2009; Ng et al, 2011; Park et al, 2009; Soon et al, 2012). The active-site cleft of PP2C is covered by the ABA-bound PYL, using a hydrophobic pocket, to prevent access to substrates of PP2Cs (Miyazono et al, 2009; Yin et al, 2009). The second activation step is executed by Raf-like protein kinases (RAFs) that directly phosphorylate and activate SnRK2s (Lin et al, 2021; Lin et al, 2020; Saruhashi et al, 2015; Soma et al, 2020; Takahashi et al, 2020), among which the constantly active B2 RAFs and stress-activated B3 RAFs are essential for ABA-induced activation of SnRK2s in subgroup III (SnRK2.2/2.3/2.6) in _Arabidopsis_, while in the moss _Physcomitrella patens_, the ABA- and stress-activated B3 RAF activates PpSnRK2 (Lin et al, 2021; Saruhashi et al, 2015; Soma et al, 2023). Similarly, the osmotic stress-activated B4 RAFs phosphorylate and activate the ABA-independent SnRK2s in subgroup I (SnRK2.1/2.4/2.5/2.9/2.10) and function together with B2 and B3 RAFs for the activation of SnRK2s under osmotic stress in _Arabidopsis_ (Lin et al, 2020; Soma et al, 2020). Since B2 RAFs are constantly active and sufficient for SnRK2.2/3/6 activation (Lin et al, 2021; Soma et al, 2023), the

[1]Shanghai Center for Plant Stress Biology, CAS Center for Excellence in Molecular Plant Sciences, Chinese Academy of Sciences, 200032 Shanghai, China. [2]University of Chinese Academy of Sciences, 100049 Beijing, China. [3]Key Laboratory of Plant Carbon Capture, Chinese Academy of Sciences, 200032 Shanghai, China. [4]Department of Endocrinology, Institute of Endocrine and Metabolic Diseases, The First Affiliated Hospital of USTC, Division of Life Sciences and Medicine, University of Science and Technology of China, 230001 Hefei, Anhui, China. [5]These authors contributed equally: Guo-Jun Li, Kong Chen. ✉E-mail: yangzhao@psc.ac.cn

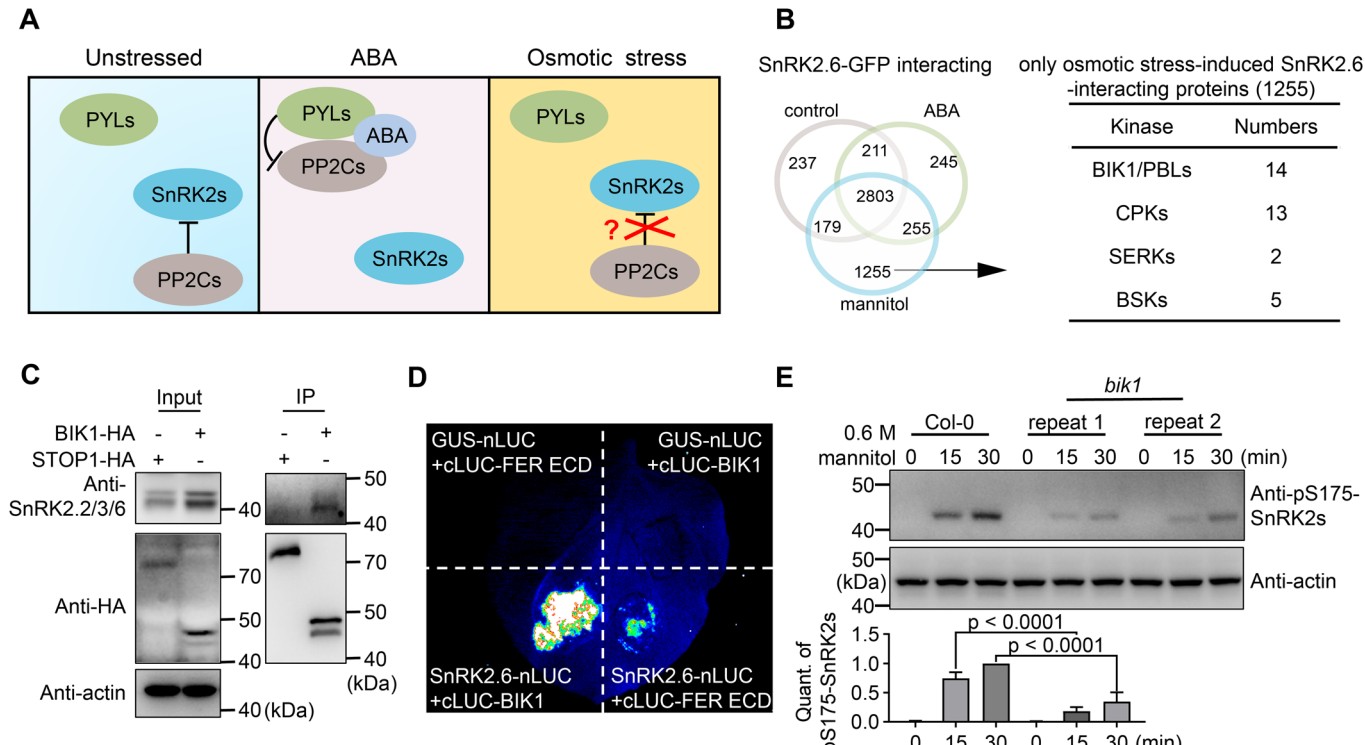

**Figure 1. BIK1 interacts with SnRK2.6 and regulates SnRK2 activation under osmotic stress.**

(A) A model illustrating the release of SnRK2s from PP2C inhibition during either ABA or osmotic stress signaling. Under unstressed conditions, clade A PP2Cs bind to and inhibit SnRK2s (left panel). In the presence of ABA, ABA receptor proteins (PYR1/PYL/RCAR or PYLs) competitively interact with PP2Cs to release subgroup III SnRK2s from PP2C-mediated inhibition (middle panel). Under osmotic stress conditions, how SnRK2s are released from PP2C-mediated inhibition remains unclear (right panel). (B) Venn diagram showing the numbers of candidate SnRK2.6-interacting proteins detected under control, 50 μM ABA, or 600 mM mannitol treatments, in extracts from *SnRK2.6pro:SnRK2.6-GFP* transgenic seedlings (left panel). Certain families of protein kinases were enriched among the SnRK2.6-GFP interacting proteins detected only under osmotic stress (right panel). The immunoprecipitation-mass spectrometry was conducted one time as a screen. (C) BIK1 interacts with endogenous SnRK2.6 in *Arabidopsis* transgenic seedlings. A co-immunoprecipitation assay with an anti-HA antibody was performed using total proteins extracted from *pBIK1:BIK1-HA* and *pSTOP1:STOP1-HA* transgenic seedlings. Immunoblotting was performed with anti-HA, anti-SnRK2.2/3/6, and anti-actin antibodies. The STOP1-HA protein was used as the negative control. (D) BIK1 interacts with SnRK2.6 in *N. benthamiana* leaves using split luciferase (LUC) complementation assay. GUS-nLUC and cLUC-FER ECD were used as negative controls. (E) Osmotic stress-induced SnRK2 activation was reduced in the *bik1* mutant. Phosphorylation of SnRK2s induced by 0.6 M mannitol was detected with the anti-phospho-S175-SnRK2s antibody (top), using total proteins extracted from wild-type (WT) Col-0 and *bik1* mutant seedlings. Actin was used as the loading control (bottom). Quantification of pS175-SnRK2s represents the band intensity ratio of phosphorylated SnRK2s to that of actin, normalized by the value of the mannitol-treated WT at 30 min. Data represent mean ± SD ($n = 3$). One-way ANOVA followed by Tukey's test. All experiments were repeated at least three times with similar results, except the immunoprecipitation-mass spectrometry (IP-MS). Source data are available online for this figure.

release of PP2C inhibition should be triggered by osmotic stress, however, it remains unclear how the PP2C-mediated inhibition of SnRK2s is released under osmotic stress.

The release of SnRK2s from PP2C inhibition can be achieved by either PP2C-interacting proteins or SnRK2-interacting proteins. Although proteins in the PYL family interact with PP2Cs in the absence of ABA (Hao et al, 2011; Li et al, 2013; Sun et al, 2019; Zhao et al, 2013), activation of SnRK2s by osmotic stress is not blocked in the *pyl* duodecuple mutant, but rather is enhanced (Zhao et al, 2018b), suggesting disinhibition of SnRK2s from PP2Cs by osmotic stress is independent of PYLs. The ABA-Insensitive (ABI) *abi1-1* mutant causes a substitution mutation (G180D) in the PP2C catalytic domain that impairs PYL binding and leads to constitutive interaction of PP2C with and inactivation of SnRK2s (Ma et al, 2009; Park et al, 2009; Umezawa et al, 2009), suggesting that the ABI1[G180D] mutation disrupts the ABA- and PYL-mediated release of SnRK2s. Although the *abi1-1* mutant is defective in SnRK2 activation by ABA, activation of SnRK2s by osmotic stress is

not altered (Vlad et al, 2010; Yoshida et al, 2006), suggesting that ABI1[G180D] does not interfere with the osmotic signaling-mediated release of SnRK2s. Moreover, ABA signaling activates SnRK2.2/2.3/2.6/2.7/2.8, while osmotic signaling activates all SnRK2s except SnRK2.9 (Boudsocq et al, 2004), suggesting that the two signaling pathways may differ in targeting negative regulators such as PP2Cs. Therefore, osmotic signaling releases PP2C-mediated inhibition of SnRK2s in a novel manner, probably via unknown PP2C- and/or SnRK2-interacting proteins (Fig. 1A, right panel).

Although the osmosensing mechanism is still unclear in plants, osmotic sensing and early signaling occur at the plasma membrane (Yu et al, 2024a). Hyperosmotic stress increases local membrane tension surrounding the cell wall adhesions, which may be sensed by the mechanically activated ion channel Reduced Hyperosmolality-Induced [$Ca^{2+}$]$_i$ Increase 1 (OSCA1) and triggers $Ca^{2+}$ signals in plants (Qin et al, 2024; Yu et al, 2024b; Yuan et al, 2014). The plasma membrane-localized phospholipid-binding BONZAI (BON) proteins are critical for osmotic stress signaling

and positively mediate diverse osmotic stress responses, including $Ca^{2+}$ signals, ABA accumulation, reprogramming of gene expression, and plant growth maintenance (Chen et al, 2020). Moreover, the plasma membrane-localized Rho-of-Plant 6 (ROP6) [a guanosine triphosphatase (GTPase)] forms nanoclusters quickly with the NADPH oxidases BURST OXIDASE HOMOLOG PROTEIN D (RBOHD) and RBOHF under osmotic stress and regulates the production of reactive oxygen species (ROS) (Martiniere et al, 2019; Smokvarska et al, 2020). The dynamics but not the formation of ROP6 nanoclusters requires the kinase FERONIA (FER) [a *Catharanthus roseus* receptor-like kinase 1-like (*Cr*RLK1L)] and the anionic lipid phosphatidylserine (PS) at the plasma membrane (Smokvarska et al, 2023). Physiological studies suggest that osmotic stress-induced ABA accumulation requires an intact cell wall (Bacete et al, 2022) and that the peptide CLAVATA 3 (CLV3)/EMBRYO-SURROUNDING REGION-RELATED 25 (CLE25) modulates ABA production via its receptors BARELY ANY MERISTEM (BAM) 1/3 in *Arabidopsis* under drought (Takahashi et al, 2018). It could be deduced that cell surface signaling may be involved in the releasement of SnRK2s under osmotic stress and that the PP2C- or SnRK2-interacting proteins may transduce the cell surface signals to the intracellular SnRK2s.

Here, we report the critical role of the receptor-like cytoplasmic kinase (RLCK) BOTRYTIS-INDUCED KINASE 1 (BIK1) in osmotic stress signaling. BIK1 localizes to the plasma membrane and nucleus and has a critical role in transducing cell surface signals to intracellular events during immune responses (Bi et al, 2018; Lal et al, 2018; Lu et al, 2010; Monaghan et al, 2014; Zhang et al, 2010). We report here that BIK1 interacts with and phosphorylates SnRK2.6 and that this phosphorylation releases the PP2C-mediated inhibition. The *bik1* mutant is defective in diverse osmotic stress responses, including SnRK2 activation, ABA accumulation, expression of stress-responsive genes, control of water loss, and maintenance of plant growth. The BIK1-mediated tyrosine phosphorylation of SnRK2.6 is required for SnRK2.6 kinase activity and subsequent osmoregulation. Our findings reveal a unique mechanism through which SnRK2s are released from inhibition during osmotic signaling in higher plants and uncover a critical role of BIK1 in osmotic stress responses.

# Results

## BIK1 interacts with SnRK2.6 and regulates SnRK2 activation under osmotic stress

SnRK2 are evolutionarily conserved kinases that originated in algae and are key regulators of the response to osmotic stress. Before SnRK2s can be activated, the constant interactions between PP2Cs and SnRK2s need to be disrupted, which can be done by PP2C- and SnRK2-interacting proteins. Previous analyses of PP2C-interacting proteins revealed that the PYL/RCAR ABA receptors can release SnRK2s from PP2C-mediated inhibition in the presence of ABA (Ma et al, 2009). The SnRK2 that has the strongest interaction with PP2C is SnRK2.6 [also named OPEN STOMATA 1 (OST1), or SRK2E], which is also a protein kinase of central importance for elucidating the osmotic stress signaling mechanism (Boudsocq et al, 2004; Fujii et al, 2011; Fujii and Zhu, 2009; Fujita et al, 2009; Lind et al, 2015; Nakashima et al, 2009; Shinozawa et al, 2019; Soon et al,

2012). We used immunoprecipitation (IP) followed by liquid chromatography-tandem mass spectrometry (LC–MS/MS) to identify *Arabidopsis* proteins interacting with SnRK2.6 using *SnRK2.6pro:SnRK2.6-GFP* transgenic plants treated with 600 mM mannitol or 50 μM ABA for 30 min. We detected 1255 proteins that may interact with SnRK2.6 in osmotic stress-treated samples but not in control or ABA-treated samples (Fig. 1B and Dataset EV1). The predicted proteins include some which are known to control early osmotic stress signaling or osmoregulation, such as three OSCA1-type $Ca^{2+}$ channels (OSCA1.1/1.2/1.5) (Yuan et al, 2014), eight Autoinhibited $Ca^{2+}$-ATPases (ACA1/2/4/8/9/10/11/13) (Yang et al, 2017), the phospholipid-binding protein BON2 (Chen et al, 2020), two *Cr*RLK1Ls (Smokvarska et al, 2023), and the sucrose transporter SWEET11 (Chen et al, 2022). Interestingly, 14 receptor-like cytoplasmic kinases in the BIK1/PBS1-LIKE (PBL) family and 13 members of the $Ca^{2+}$-dependent protein kinase (CDPK or CPK) family were identified as putative SnRK2-interacting proteins under osmotic stress. This high number of kinases from these two families suggests their central roles in regulating SnRK2.6 under osmotic stress. Using transgenic seedlings developed in another study (*super:OST1-Myc*) (Ding et al, 2015), we detected 686 proteins by LC–MS/MS that may interact with SnRK2.6/OST1 in osmotic stress-treated samples but not in control samples (Fig. EV1A; Dataset EV2), among which there were 5 BIK1/PBL proteins (Fig. EV1A).

BIK1 and PBLs belong to the RLCK subfamily VII and are critical in transducing cell surface signals to intracellular events (Bi et al, 2018; Lal et al, 2018; Lu et al, 2010; Monaghan et al, 2014; Zhang et al, 2010), which may also occur in osmotic stress signaling. We further confirmed physical interactions between BIK1 and SnRK2.6 using co-IP and split luciferase (LUC) complementation (LCI) assays. The endogenous SnRK2.2/2.3/2.6 were co-immunoprecipitated with BIK1-HA from *pBIK1:BIK1-HA* transgenic plants (Fig. 1C), as detected by an anti-SnRK2.2/3/6 antibody. Moreover, the transient co-expression of *SnRK2.6-nLUC* and *cLUC-BIK1* generated strong reconstituted LUC signals in *Nicotiana benthamiana* (*Nb*) leaves (Fig. 1D). To determine the subcellular localization of the BIK1-SnRK2.6 interaction, we transiently coexpressed *BIK1-YFP^C* and *SnRK2.6-YFP^N* with the plasma membrane marker *PIP2-mRFP* in *Nicotiana benthamiana* (*Nb*) leaves and found that the interaction mainly occurs at the plasma membrane (Fig. EV1B). These data verify that BIK1 is an SnRK2.6-interacting protein.

Release of SnRK2s from PP2C-mediated inhibition is required for any subsequent activation of SnRK2s. To examine whether BIK1 participates in SnRK2 activation, we detected SnRK2 activity using western blot analysis with anti-phospho-Ser175-SnRK2.6 antibody, which recognizes the phosphorylated serine residue in the activation loop of SnRK2.2/3/6 (Zhao et al, 2018b). Indeed, phosphorylation of SnRK2s induced by mannitol, but not by ABA, was reduced in the T-DNA knockout *bik1* mutant compared to the wild-type (WT) (Figs. 1E and EV1C–F), suggesting that BIK1 may have the ability to disrupt PP2C-SnRK2 interactions or to activate SnRK2s under osmotic stress.

## BIK1 releases SnRK2.6 from PP2C binding and inhibition

Since BIK1 interacts with and activates SnRK2.6, we were motivated to examine whether BIK1 interferes with PP2C binding

## A

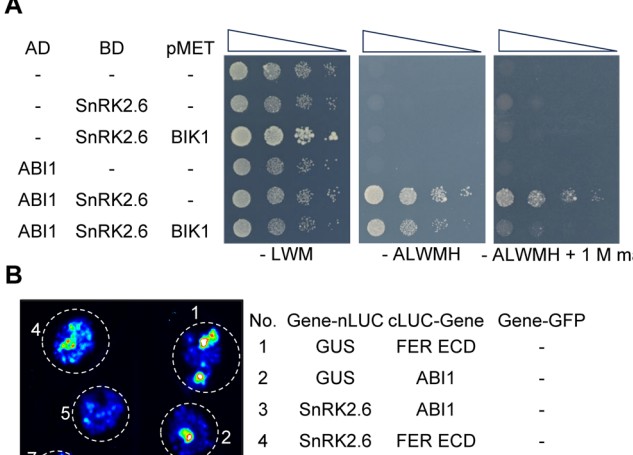

## B

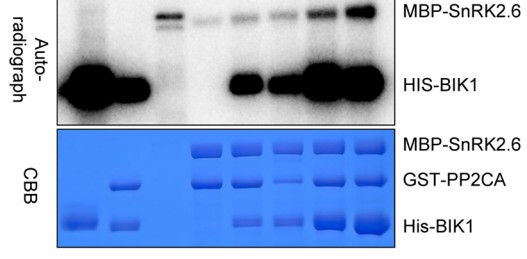

| No. | Gene-nLUC | cLUC-Gene | Gene-GFP |
|---|---|---|---|
| 1 | GUS | FER ECD | - |
| 2 | GUS | ABI1 | - |
| 3 | SnRK2.6 | ABI1 | - |
| 4 | SnRK2.6 | FER ECD | - |
| 5 | SnRK2.6 | ABI1 | 1 OD BIK1 |
| 6 | SnRK2.6 | ABI1 | GFP |
| 7 | SnRK2.6 | ABI1 | 0.1 OD BIK1 |

## C

**Figure 2. BIK1 releases SnRK2.6 from PP2C-mediated interaction and inhibition.**

(A, B) SnRK2.6-ABI1 interaction is disrupted by BIK1, as evaluated by yeast three-hybrid (Y3H) (A) and split LUC complementation assays (B). For Y3H assays, transformed yeast cells were grown on nonselective medium lacking Leu, Trp, and Met (SD/ − LWM), and the selective medium lacking Ade, Leu, Trp, Met, and His (SD/ − ALWMH), without or with 1 M mannitol (A). The AD-ABI1 and BD-SnRK2.6 combination was used as a positive control. For split LUC complementation assays, SnRK2.6 and ABI1 were fused to the split N- or C-terminal fragments of LUC (SnRK2.6-nLUC and cLUC-ABI1). The SnRK2.6-nLUC and cLUC-ABI1 combination was used as a positive control, and combinations of SnRK2.6-nLUC/cLUC-FER ECD, GUS-nLUC/cLUC-ABI1, and SnRK2.6-nLUC/cLUC-ABI1/GFP were used as negative controls. (C) BIK1 releases and activates SnRK2.6 from PP2CA-mediated inhibition during in vitro phosphorylation assays. The MBP-SnRK2.6 kinase was pre-inactivated by incubation with GST-PP2CA for 30 min and then re-activated in the presence of His-BIK1 protein for one hour. The phosphorylation of MBP-SnRK2.6 was detected by autoradiography (upper panel). Protein loading was detected by Coomassie blue staining (CBB, lower panel). All experiments were repeated at least three times with similar results. Source data are available online for this figure.

and inhibition of SnRK2s. The SnRK2.6-ABI1 complex was selected to test the potential role of BIK1 in releasing SnRK2s, because of the evolutionarily conserved function of this SnRK2-PP2C complex in osmoregulation and its strong interaction among the evaluated SnRK2-PP2C complexes (Komatsu et al, 2013; Lind et al, 2015;

Rubio et al, 2009; Soon et al, 2012; Tougane et al, 2010; Umezawa et al, 2009). We conducted yeast three-hybrid (Y3H) and LCI assays and found that the SnRK2.6-ABI1 interaction was reduced by co-expression of BIK1 (Fig. 2A,B). In Y3H assays, we fused SnRK2.6 to the GAL4 DNA-binding domain (BD) and ABI1 to the GAL4-activating domain (AD). Interactions were detected between ABI1 and SnRK2.6 (Fig. 2A). Co-expression of BIK1 slightly repressed the SnRK2.6-ABI1 interaction under unstressed conditions, and this inhibition was not caused by altered expression of SnRK2.6 or ABI1 (Fig. EV2A). When the yeasts were grown on media with 1 M mannitol, BIK1 dramatically repressed the SnRK2.6-ABI1 interaction (Figs. 2A and EV2A), suggesting that BIK1 releases SnRK2.6 from PP2C binding and that its activity may be elevated by osmotic stress through an unknown mechanism in yeast. In the LCI assay, the transient co-expression of SnRK2.6-nLUC and cLUC-ABI1 generated strong reconstituted LUC signals in Nb leaves. The SnRK2.6-ABI1 interaction was dramatically repressed by the co-expression of BIK1-GFP, but not the negative control, GFP alone (Fig. 2B). Moreover, a lower dose of BIK1-GFP caused a milder inhibition of the SnRK2.6-ABI1 interaction, and BIK1-GFP also repressed the SnRK2.6-HAB1 interaction in the LCI assay (Figs. 2B and EV2B; HAB1 is another PP2C). These results indicated that BIK1 can break the interaction between SnRK2.6 and PP2Cs.

We have now shown that BIK1 can interact with SnRK2.6 (Figs. 1C,D and EV1B) and can break the interaction between SnRK2.6 and PP2Cs (Figs. 2A,B and EV2B). We next investigated whether BIK1 also interacts with ABI1 using yeast two-hybrid (Y2H) and LCI assays. We fused BIK1 to the GAL4 DNA-binding domain (BD) and ABI1 to the GAL4-activating domain (AD), but did not detect any interaction between BIK1 and ABI1 (Fig. EV2C). In Nb leaves, the pairing of BIK1-nLUC with cLUC-ABI1 did not generate strong reconstituted LUC signals (Fig. EV2D). Thus, our results suggest that BIK1 breaks the interaction between SnRK2.6 and PP2Cs not through long-term interaction with the PP2Cs but through regulating SnRK2.6.

To examine whether BIK1 releases SnRK2s from PP2C-mediated inhibition, we conducted in vitro kinase assays using recombinant kinases and phosphatases. After pre-incubating MBP-SnRK2.6 and GST-PP2CA in protein kinase buffer, different concentrations of His-BIK1 were added to reactions and incubated. MBP-SnRK2.6 and His-BIK1 protein kinases showed autophosphorylation activity, and incubation of GST-PP2CA substantially decreased SnRK2.6 phosphorylation (Fig. 2C). The PP2CA-mediated inhibition of SnRK2.6 phosphorylation was gradually released by increasing concentrations of BIK1 (Fig. 2C). These results suggested that BIK1 not only breaks the interaction between SnRK2.6 and PP2Cs but also releases the PP2C inhibition of SnRK2.6. Another possibility is that BIK1 may phosphorylate and activate SnRK2.6 regardless of the inhibition by PP2CA (Fig. 2C).

## Release of SnRK2s from PP2Cs by BIK1 differs from that of PYLs

During ABA- and PYL-mediated signaling, the ABA-bound PYL receptors bind the ABI1 phosphatase to release SnRK2 from ABI1-mediated inhibition. Since disinhibition of SnRK2s by osmotic stress is independent of PYLs and is not disrupted by the ABA-insensitive ABI1$^{G180D}$ mutation (Vlad et al, 2010; Zhao et al, 2018b)

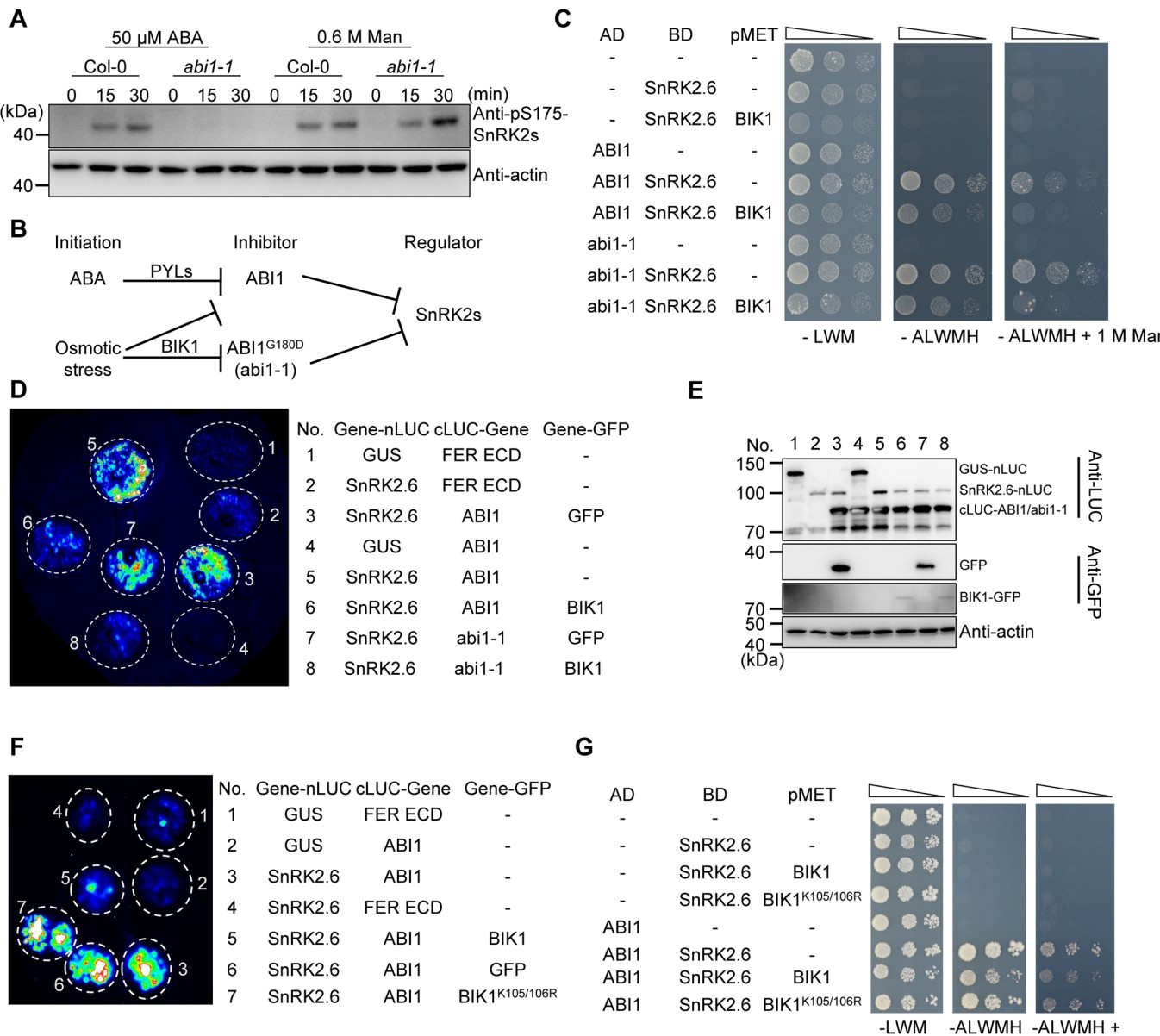

(Fig. 3A,B), we next examined whether ABI1$^{G180D}$ disrupts the BIK1-mediated release of SnRK2.6. Using Y3H and LCI assays, we found that co-expression of BIK1 repressed both the SnRK2.6-ABI1 and SnRK2.6-ABI1$^{G180D}$ interactions (Fig. 3C–E). First, we detected interactions between SnRK2.6 and either wild-type ABI1 or ABI1$^{G180D}$ in the Y3H assay. Co-expression of BIK1 repressed both interactions, especially when grown on media with 1 M mannitol (Fig. 3C). Second, the pairing of SnRK2.6 with wild-type ABI1 or mutated ABI1$^{G180D}$ caused strong reconstituted LUC signals in *Nb* leaves, and this LUC signal was dramatically repressed by the co-expression of BIK1-GFP (Fig. 3D,E). These results suggested that BIK1-mediated release of SnRK2.6 differs from the well-characterized PYL-mediated release and utilizes a unique SnRK2-PP2C interface that is not affected by the ABI1$^{G180D}$ mutation.

We next investigated whether BIK1-mediated disruption of the SnRK2.6-ABI1 interaction requires BIK1 kinase activity. We expressed the BIK1$^{K105/106R}$, a catalytically inactive variant of BIK1 (Lu et al, 2010; Monaghan et al, 2014), in Y3H and LCI assays. Interestingly, the SnRK2.6-ABI1 interaction was strongly repressed by co-expression of wild-type BIK1 but not the kinase-dead BIK1$^{K105/106R}$, as shown by reconstituted LUC signals in *Nb* leaves (Fig. 3F) and yeast growth on synthetic dextrose (SD) media (−Ade, −Leu, −Trp, −Met, and −His) with 1 M mannitol (Fig. 3G). These results indicated that BIK1 kinase activity is essential for disrupting the interaction between SnRK2.6 and ABI1.

## Two tyrosine residues phosphorylated by BIK1 are critical for ABI1–SnRK2.6 interaction

Since BIK1 is a nonreceptor dual-specificity kinase with both serine/threonine and tyrosine autophosphorylation activities (Lin et al, 2014), and its kinase activity is required for releasing

◀ **Figure 3. BIK1-mediated release of SnRK2s from PP2Cs differs from its release by PYLs.**

(A) The abi1-1 (ABI1^G180D) mutant is defective in SnRK2 activation by ABA but not by osmotic stress. Phosphorylation of SnRK2s induced by 50 μM ABA or 0.6 M mannitol was detected with the anti-phospho-S175-SnRK2s antibody (top), using total proteins extracted from 9-day-old WT Col-0 and abi1-1 mutant seedlings. Actin was used as the loading control (bottom). (B) A model illustrating the difference in the release of SnRK2s from PP2C inhibition by ABA and osmotic stress in the abi1-1 mutant. The dominant ABI1^G180D mutation disrupts its interaction with PYLs and disables PYL-mediated release of SnRK2s. Therefore, the interaction between ABI1^G180D (abi1-1) and SnRK2s cannot be disrupted by PYLs in the presence of ABA. The ABI1^G180D mutation did not affect the activation of SnRK2s by osmotic stress. (C, D) BIK1 disrupts the interaction between SnRK2.6 and ABI1^G180D in Y3H (C) and split LUC complementation assays (D). For Y3H assays, transformed yeast cells were grown on a nonselective SD/ − LWM medium, and the selective SD/ − ALWMH medium, without or with 1 M mannitol. The combinations of AD-ABI1/BD-SnRK2.6 and AD-abi1-1/BD-SnRK2.6 were used as positive controls (C). For split LUC complementation assays, SnRK2.6 and ABI1 or ABI1^G180D were fused to the split N- or C-terminal fragments of LUC (SnRK2.6-nLUC and cLUC-ABI1 or cLUC-abi1-1). The SnRK2.6-nLUC and ABI1-cLUC combination was used as a positive control, and combinations of SnRK2.6-nLUC/ cLUC-FER ECD, GUS-nLUC/cLUC-ABI1, and SnRK2.6-nLUC/cLUC-ABI1/GFP were used as negative controls (D). (E) Protein expression levels of SnRK2.6, ABI1 and BIK1 in Nicotiana benthamiana leaves for split LUC complementation assays. Total proteins were extracted from the leaves expressing GUS-nLUC, SnRK2.6-nLUC, cLUC-ABI1, cLUC-ABI1^G180D, cLUC-FER ECD, GFP, and BIK1-GFP (as numbered in (D)), and detected by western blot using anti-LUC and anti-GFP antibodies. The anti-actin antibody was used as a loading control. The resulting LUC signals in the split LUC complementation assay are shown in Fig. 3D. (F, G) BIK1 kinase activity is required for the disruption of SnRK2.6-ABI1 interaction in Y3H (F) and split LUC complementation assays (G). For split LUC complementation assays, the combinations of SnRK2.6-nLUC/ cLUC-ABI1 and SnRK2.6-nLUC/cLUC-ABI1/GFP were used as positive controls, and combinations of SnRK2.6-nLUC/cLUC-FER ECD and GUS-nLUC/cLUC-ABI1 were used as negative controls (F). For Y3H assays, transformed yeast cells were grown on a nonselective SD/− LWM medium, and the selective SD/ − ALWMH medium, without or with 1 M mannitol. The AD-ABI1 and BD-SnRK2.6 combination was used as a positive control (G). All experiments were repeated at least three times with similar results. Source data are available online for this figure.

SnRK2.6, we speculated that BIK1 may regulate SnRK2.6-PP2C interactions through phosphorylation. We detected no phosphorylation of PP2CA or ABI1 by BIK1 (Figs. 2C and EV3A), thus we examined whether BIK1 phosphorylates SnRK2.6 using in vitro kinase assays. The catalytically inactive variants of SnRK2.4 (K33N) and SnRK2.6 (K50N) exhibited no autophosphorylation activities; however, MBP–BIK1 exhibited strong autophosphorylation and transphosphorylation activities on GST-SnRK2.6^K50N (Fig. EV3B). Moreover, MBP–BIK1 phosphorylated GST-SnRK2.2^K52N and GST-SnRK2.3^K51N in vitro (Fig. EV3C).

To systematically examine BIK1-mediated transphosphorylation on SnRK2.6, we performed LC–MS/MS after in vitro phosphorylation reactions of SnRK2.6^K50N by BIK1 using ^18O-ATP as the phosphate donor. We identified 13 putative BIK1-mediated phosphosites. There were eight serine residues (Ser, S), three threonine residues (Thr, T), and two tyrosine residues (Tyr, Y) in SnRK2.6^K50N (S29, S43, S113, T146, Y163, S164, S166, S167, S171, S175, T176, T179, and Y182) (Fig. EV3D), among which six S/T residues, namely S29, S43, S171, S175, T176, and T179, were also identified as putative autophosphorylation sites of the serine/ threonine kinase SnRK2.6 (Fig. EV3E) (Lin et al, 2021).

We next examined whether the phospho-mimic and non-phosphorylatable versions of SnRK2.6 showed altered interactions with ABI1 using Y2H and LCI assays (Figs. 4A and EV3F). The phospho-mimic (S/T/Y-to-D) mutations of all 13 putative BIK1-mediated phosphosites (SnRK2.6^13Ds) strongly reduced the SnRK2.6 and ABI1 interaction in the LCI assay (Fig. 4A). The non-phosphorylatable (S/T/Y-to-A/F) mutations (SnRK2.6^13A/Fs) exhibited similar interaction with ABI1 compared with that of wild-type SnRK2.6 both in LCI and Y2H assays (Figs. 4A and EV3F). The BD-SnRK2.6^13Ds fusion showed strong self-activation in the Y2H assay and did not exhibit improved growth in the presence of AD-ABI1 (Figs EV3F). These results further support that BIK1 disrupts the SnRK2.6-ABI1 interaction through phosphorylation regulation.

Because SnRK2.6 cannot release itself from PP2C-mediated interaction and inhibition, we speculated that the autophosphorylation sites are not critical for disrupting the interactions between SnRK2.6 and PP2Cs. Since BIK1 has tyrosine autophosphorylation activity (Lin et al, 2014), we focused on elucidating the roles of the

two tyrosine residues. The two tyrosine residues in SnRK2.6 are adjacent to the interface between SnRK2.6 and HAB1 (Fig. EV3G).

Tyrosine phosphorylation was detected using in vitro kinase assays with (γ-^32P) ATP (Fig. 4B) or following western blot analysis with the anti-phospho-tyrosine antibody (Fig. 4C). We mutated each tyrosine in SnRK2.6^K50N to a similar-sized, non-phosphorylatable phenylalanine (F). We found that the single mutations Y163F and Y182F reduced the phosphorylation of SnRK2.6^K50N and that the double mutation in SnRK2.6^K50N/Y163F/Y182F nearly abolished the BIK1-mediated tyrosine phosphorylation (Fig. 4B,C). These two highly conserved tyrosine residues (Y163 and Y182) are located in the activation loop of the SnRK2s (Figs. 4D and EV3H). Structural analyses of SnRK2.6 and ABI1 or HAB1 indicate that Y182 is spatially close to where the conserved tryptophan of the PP2Cs (which also binds ABA) docks (W385 in HAB1, W300 in ABI1, excluding AHG1) (Melcher et al, 2009; Miyazono et al, 2009; Soon et al, 2012; Yin et al, 2009) (Figs. 4E,F and EV3G). The tryptophan lock of PP2Cs inserts into both SnRK2.6 and the PYR/PYL receptors, and the insertion of tryptophan lock into the PYL receptors is essential for the inhibition of PP2C by PYL (Melcher et al, 2009; Miyazono et al, 2009; Soon et al, 2012; Yin et al, 2009). Residue Tyr163 in the SnRK2s is near the Mg^2+ binding Asp-Phe-Gly (DFG) motif (Fig. EV3H). Structural simulation suggests that the phosphorylation of the conserved tyrosine increases the hydrophilicity and negative charges of the SnRK2.6 catalytic cleft and repels the docking of the nonpolar tryptophan lock of ABI1 (Fig. 4G). The Y182F and Y163F mutations in the MBP-tagged SnRK2.6 preserved SnRK2 autophosphorylation and transphosphorylation activities; although inhibition of SnRK2.6 by ABI1 and PP2CA was reduced by the Y163F mutation, especially in PP2CA-mediated SnRK2.6 inhibition (Figs. 4H,I and EV4A,B). Therefore, the phosphorylation of these two conserved tryptophan residues in SnRK2.6 may alter the inhibition by PP2Cs.

## The tryptophan lock of PP2C plays a critical role in PP2C-SnRK2.6 binding

According to structural evidence, the tryptophan lock of PP2Cs is central for SnRK2.6 binding. However, triple mutation of the tryptophan lock did not significantly weaken the HAB1-SnRK2.6

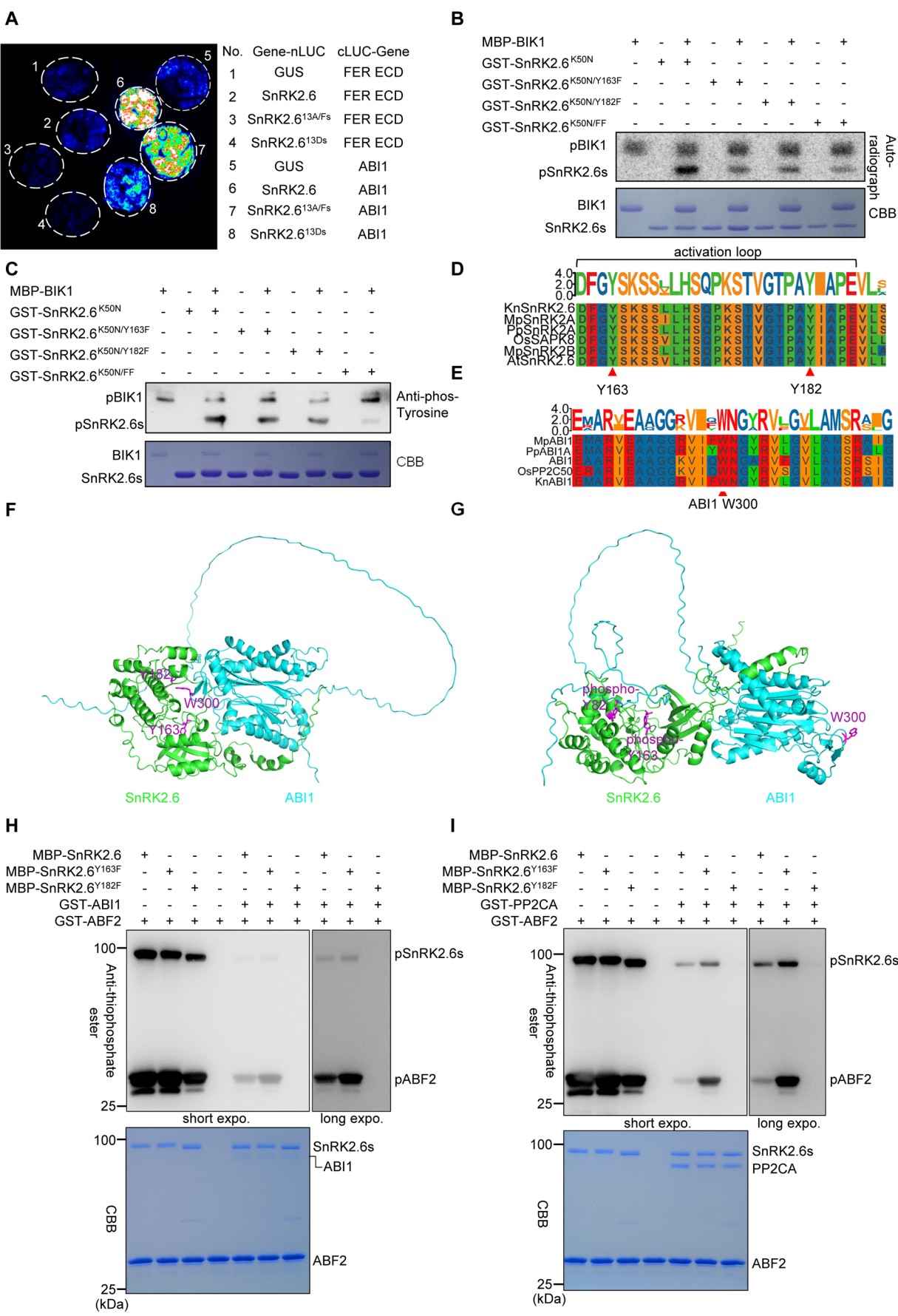

**Figure 4. Two tyrosine residues phosphorylated by BIK1 are important for the ABI1–SnRK2.6 interaction.**

(A) The phospho-mimic mutations (S/T/Y to D) of all 13 putative BIK1-mediated phosphosites (SnRK2.6$^{13Ds}$) strongly reduced the SnRK2.6 and ABI1 interaction in the split LUC complementation assay in *Nicotiana benthamiana* leaves. In contrast, the non-phosphorylatable mutations (S/T/Y to A/F) of the putative BIK1-mediated phosphosites (SnRK2.6$^{13A/Fs}$) exhibited similar interaction with ABI1 compared with that of wild-type SnRK2.6. The combination of SnRK2.6-nLUC/cLUC-ABI1 was used as a positive control, and combinations of SnRK2.6-nLUC/cLUC-FER ECD and GUS-nLUC/cLUC-ABI1 were used as negative controls. (B, C) Phosphorylation of catalytically inactive SnRK2.6$^{K50N}$, without or with the Y-to-F mutations [Y163F, Y182F, and Y163F/Y182F (FF)], by recombinant MBP–BIK1 during in vitro phosphorylation assays. (B) Autoradiography (top) and CBB staining (bottom) exhibited phosphorylation and loading of GST-SnRK2.6$^{K50N}$ variants and MBP–BIK1, respectively. (C) The anti-phospho-tyrosine antibody was used to detect tyrosine phosphorylation (top), and CBB staining (bottom) exhibited the loading of proteins. (D) Tyr163 and Tyr182 sites of SnRK2.6 are conserved across plant species. The conserved tyrosines are marked with red triangles. *Kn, Klebsormidium nitens; Mp, Marchantia polymorpha; Pp, Physcomitrella patens; Os, Oryza sativa*. (E) The Trp300 sites of ABI1 are conserved across plant species. The conserved Trp is indicated by a red triangle. (F, G) Cartoon presentation of the SnRK2.6 and ABI1 interactions, without (F) or with (G) modeling phosphorylation by AlphaFold3, suggests phosphorylation of the two tyrosines can increase the hydrophilicity and negative charges of the SnRK2.6 catalytic cleft and repel the docking of the nonpolar tryptophan lock of ABI1. The key tyrosine residues (magenta, Y163 and Y182) in SnRK2.6 (green) are adjacent to the tryptophan lock (magenta, W300) of ABI1 (cyan) in unphosphorylated form (F), but not phosphorylated form (G). (H, I) The Y163F mutation of SnRK2.6 reduces the inhibition of SnRK2.6 by ABI1 (H) and PP2CA (I) during in vitro phosphorylation assays using the ATP analog ATPγS. After the PNBM alkylation reaction, the thiophosphate ester groups on the substrate were detected by the anti-thiophosphate ester antibody (top). GST-ABF2 fragment was used as the substrate for SnRK2.6. CBB staining (bottom) exhibited the loading of proteins. "Short expo" means short exposure of the membrane, while "long expo" means long exposure. All experiments were repeated at least three times with similar results. Source data are available online for this figure.

interaction as detected by the AlphaScreen luminescence proximity assay (Soon et al, 2012), which somehow contradicts the structural and biochemical evidence. The W300 residues of ABI1 are highly conserved among ABI1 proteins in plants (Fig. 4E). Therefore, we examined interference of PP2C-SnRK2.6 interactions by substitution of the nonpolar tryptophan (W) to a positively charged arginine (R), which alters the hydrophilicity of the lock, or a small nonpolar alanine (A), which eliminates protrusion of the lock. In Y2H and LCI assays, both the W300R and W300A substitutions in ABI1 dramatically reduced, but did not block, the ABI1–SnRK2.6 interactions (Fig. 5A–D; Appendix Fig. S1A–D). Likewise, the W385A substitution of HAB1 also reduced its interaction with SnRK2.6 (Fig. 5E; Appendix Fig. S1E). Moreover, the W280A substitution of PP2CA decreased the PP2CA-mediated inhibition of SnRK2.6 during in vitro phosphorylation assays (Fig. 5F), as did the W300A mutation of ABI1 (Fig. EV4C). Since these mutations did not significantly alter the protein abundance in yeast cells or *Nb* leaves (Appendix Fig. S1A–E), these results demonstrated the critical role of the tryptophan lock in PP2C-SnRK2.6 binding.

## Phosphorylation of Y182 is critical for kinase activity and function of SnRK2.6

BIK1 phosphorylates the two conserved tyrosine residues of SnRK2.6 to repress SnRK2.6-ABI1 interaction (Figs. 2–4), and these two tyrosine residues are adjacent to the binding site of the tryptophan lock of ABI1 that is critical for SnRK2.6-ABI1 interactions (Fig. 5). Since the two tyrosine residues are close to the catalytic cleft, we next investigated whether tyrosine phosphorylation of SnRK2.6 interferes with its kinase activity. We found that Y163F preserved the autophosphorylation ability of SnRK2.6 and did not alter the transphosphorylation of ABF2. On the other hand, both the Y182F single and the Y163F/Y182F double mutation of GST-tagged SnRK2.6 nearly blocked both autophosphorylation of SnRK2.6 and transphosphorylation of ABF2 by SnRK2.6 (Figs. 6A and EV5A), suggesting phosphorylation of Y182 is also critical for SnRK2 activity in vitro.

To investigate whether phosphorylation of the two tyrosine residues affects SnRK2.6 function in vivo, we generated transgenic lines with native promoter-driven constructs of either a WT

SnRK2.6 or a *SnRK2.6$^{Y163F}$, SnRK2.6$^{Y182F}$*, or *SnRK2.6$^{Y163F/Y182F}$* in the *ost1-3/snrk2.6* mutant background (Fig. 6B,C). Since SnRK2.6 is a central positive regulator of stomatal closure that controls transpiration-associated water loss in response to dehydration (Mustilli et al, 2002), we analyzed water loss to evaluate the functions of wild-type and mutated SnRK2.6 in vivo. Water loss from the detached rosette was much more rapid in the *snrk2.6* mutant than the WT, which could be complemented by the *pSnRK2.6:SnRK2.6* or the *pSnRK2.6:SnRK2.6$^{Y163F}$* transgenes (Fig. 6B,C; Appendix Fig. S2). However, the accelerated water loss phenotype of the *snrk2.6* mutant could not be complemented by either *pSnRK2.6:SnRK2.6$^{Y182F}$* or *pSnRK2.6:SnRK2.6$^{Y163F/Y182F}$* transgenes (Fig. 6B,C; Appendix Fig. S2). These data suggested that phosphorylation of Y182 is required for SnRK2.6 activation and function in plants.

Within a protein, aspartate and glutamate residue substitutions cannot fully mimic the volume of phosphorylated-Tyr, but do have a similar negative charge (Hunter, 2012). Therefore, we evaluated the kinase activity of Y-to-D point mutations of SnRK2.6. Asp substitution of the two conserved Tyr residues, namely Y163D, Y182D, or Y163D/Y182D, blocked both autophosphorylation of SnRK2.6 and transphosphorylation of ABF2 (Figs. 6D and EV5B). In addition, the Y-to-E point mutations of SnRK2.6, namely Y163E, Y182E, or Y163E/Y182E, blocked both autophosphorylation of SnRK2.6 and transphosphorylation of ABF2 (Fig. EV5C). We further measured water loss from the detached rosettes of transgenic lines harboring native promoter constructs of *SnRK2.6 Y163D, Y182D*, or *Y163D/Y182D* in the *snrk2.6* background, and found that none of these *SnRK2.6* Y-to-D mutations could complement the accelerated water loss phenotype of the *snrk2.6* mutant (Fig. 6E,F; Appendix Fig, S2). Moreover, combinations of Y-to-D and Y-to-F mutations of *SnRK2.6*, namely *Y163D/Y182F* and *Y163F/Y182D*, could not complement the accelerated water loss phenotype of the *snrk2.6* mutant (Fig. EV5D; Appendix Fig. S2). Besides, structural simulation suggests that the Y-to-D and Y-to-E point mutations of SnRK2.6 can hardly mimic the phosphorylation of the two conserved Tyr residues to repel the docking of the nonpolar tryptophan lock of ABI1 (Appendix Fig. S3). These results suggest that loss of these Tyr residues may severely interfere with the catalytic cleft of SnRK2.6 and that Y163 and Y182 residues are both critical for SnRK2.6 structure and activity.

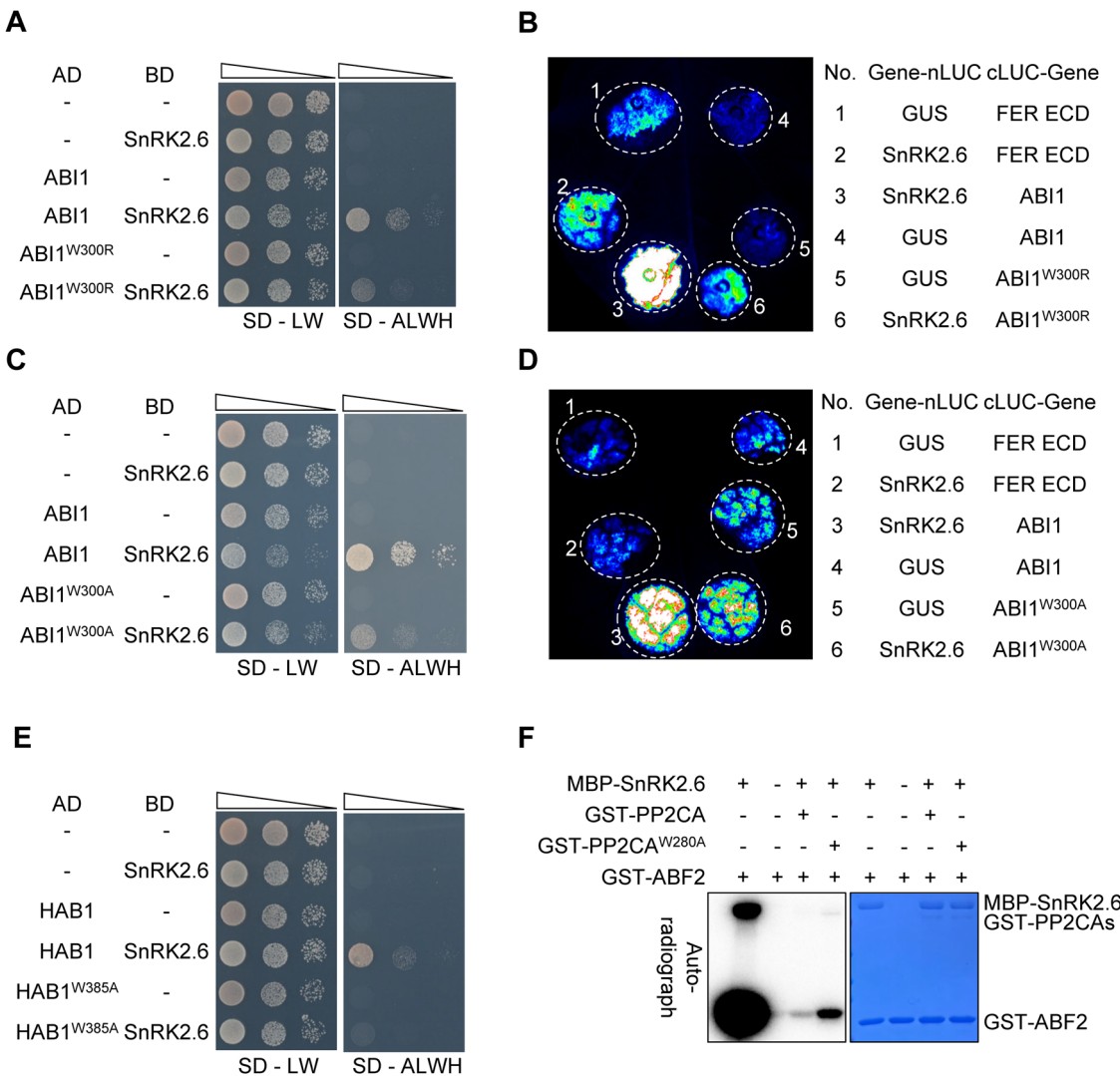

**Figure 5. A tryptophan residue of PP2C is required for PP2C-SnRK2.6 binding.**

(A–D) The tryptophan "lock" of ABI1 is required for ABI1–SnRK2.6 binding. The SnRK2.6-ABI1 interactions were dramatically reduced by the W300R (A, B) and W300A (C, D) mutations of ABI1 in Y2H (A, C) and split LUC complementation assays (B, D). The transformed yeast cells were grown on the nonselective SD/ − LW medium and the selective SD/ − ALWH medium (A, C). The AD-ABI1 and BD-SnRK2.6 combination was used as a positive control. The combination of SnRK2.6-nLUC/cLUC-ABI1 was used as a positive control, and combinations of SnRK2.6-nLUC/cLUC-FER ECD and GUS-nLUC/cLUC-ABI1 were used as negative controls. (E) The tryptophan "lock" of HAB1 is required for HAB1-SnRK2.6 binding. The SnRK2.6-HAB1 interaction was dramatically reduced by the W385A mutation of HAB1 in the Y2H assay. The transformed yeast cells were grown on the nonselective SD/ − LW medium and the selective SD/ − ALWH medium. The AD-HAB1 and BD-SnRK2.6 combination was used as a positive control. (F) The tryptophan "lock" of PP2CA is important for PP2CA-mediated inhibition of SnRK2.6. The W280A mutation of PP2CA reduced its inhibition on SnRK2.6 during in vitro phosphorylation assays. The GST-ABF2 fragment was used as the substrate for SnRK2.6. Autoradiography (left) and CBB staining (right) exhibited phosphorylation and loading of proteins, respectively. All experiments were repeated at least three times with similar results. Source data are available online for this figure.

## BIK1 controls osmotic stress responses

Since BIK1 releases SnRK2.6 from PP2C-mediated inhibition and regulates SnRK2 activation under osmotic stress, we further investigated whether BIK1 is required for diverse osmotic stress responses in plants. First, we analyzed plant growth in response to osmotic stress in the *bik1* mutant. Four-day-old seedlings of WT and *bik1* mutant of similar sizes were transferred to plates containing ½ MS or ½ MS with mannitol, sorbitol, NaCl, or ABA (Figs. 7A,B and EV1G,H; Appendix Fig. S4). After 15 days of

growth on the ½ MS control medium, the WT and *bik1* mutant had similar rosette size, root length, and fresh weight; however, the rosette growth of the *bik1* mutant was severely inhibited on the medium with mannitol (Fig. 7A,B; Appendix Fig. S4A,D), or with sorbitol and NaCl (Appendix Fig. S4B–D). Besides, the rosette growth of the *bik1* mutant was also severely inhibited on the medium with ABA (Fig. EV1G,H), suggesting that the *bik1* mutant is not defective in ABA responses. Second, we measured ABA content in 9-day-old seedlings after they were treated with mannitol and found that the *bik1* mutant had reduced ABA

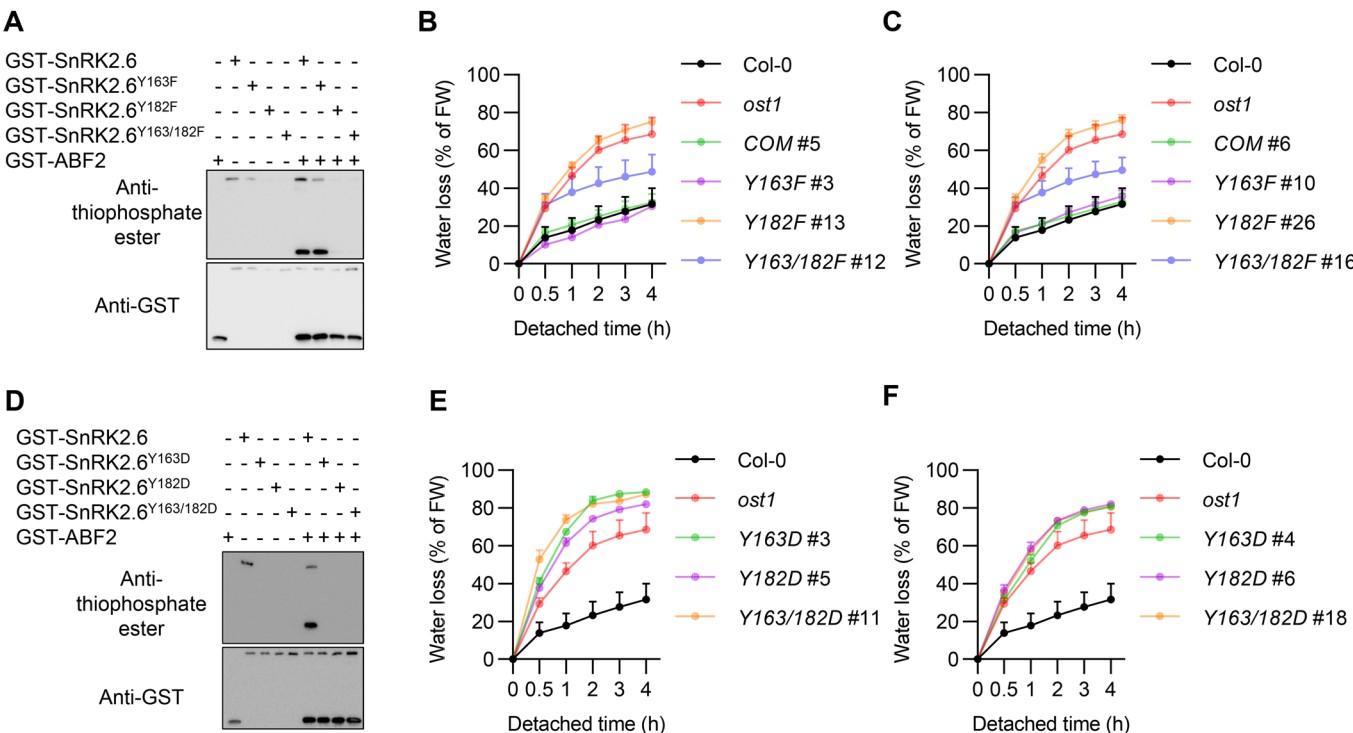

**Figure 6.  Phosphorylation of Y182 is critical for kinase activity and function of SnRK2.6.**

(A) Phosphorylation of the ABF2 fragment by wild-type and mutated SnRK2.6 proteins, namely SnRK2.6$^{Y163F}$, SnRK2.6$^{Y182F}$, SnRK2.6$^{Y163F/Y182F}$, during in vitro kinase assays using the ATP analog ATPγS. After the PNBM alkylation reaction, the thiophosphate ester groups on the substrate were detected by the anti-thiophosphate ester antibody. The anti-GST antibody was used as a loading control. (B, C) Cumulative transpirational water loss from detached rosettes of 5-week-old WT Col-0, *ost1-3* (a *snrk2.6* mutant) mutant, and transgenic plants in the *ost1-3* mutant background expressing either a wild-type or mutated *SnRK2.6*, namely *Y163F*, *Y182F*, or *Y163F/Y182F* using a native promoter (*proSnRK2.6:SnRK2.6*). Values are means ± SD ($n = 3$ independent experiments). (D) Phosphorylation of ABF2 fragment by the wild-type and mutated SnRK2.6, including SnRK2.6$^{Y163D}$, SnRK2.6$^{Y182D}$, SnRK2.6$^{Y163D/Y182D}$, during in vitro kinase assays using the ATP analog ATPγS. After the PNBM alkylation reaction, the thiophosphate ester groups on the substrate were detected by the anti-thiophosphate ester antibody. The anti-GST antibody was used as a loading control. (E, F) Cumulative transpirational water loss from detached rosettes of 5-week-old WT Col-0, *ost1-3* mutant, and *proSnRK2.6::SnRK2.6* transgenic plants in the *ost1-3* mutant background expressing either wild-type or mutated *SnRK2.6*, namely *Y163D*, *Y182D*, or *Y163D/Y182D*. Values are means ± SD ($n = 3$ independent experiments). All experiments were repeated at least three times with similar results. Water loss for the Col-0 wild-type and the *ost1* mutant was repeated 12 times and was shared by (B, C, E, F). Source data are available online for this figure.

accumulation compared with WT (Fig. 7C). Third, we analyzed stress-responsive gene expression and found that the *bik1* mutant had reduced expression of the key marker genes *RAB18* and *KIN1* (Fig. 7D). Fourth, we evaluated transpiration-associated water loss in detached rosettes and found that the *bik1* mutants exhibited accelerated water loss compared to WT in 5-week-old plants (Fig. 7E). Finally, we conducted an in vitro kinase assay with immunoprecipitated BIK1-HA from the *pBIK1:BIK1-HA* transgenic line and found that BIK1 was strongly activated after 15 min of treatment with 600 mM mannitol (Fig. 7F). We further confirmed osmotic stress-induced BIK1 activation with *35S:BIK1-Myc* transgenic line, with flg22 treatment as a positive control (Fig. 7G). ABA signaling is critical for plant growth and stress responses (Zhao et al, 2018b). To evaluate whether BIK1 functions redundantly with ABA in mediating SnRK2 release and plant growth (Fig. 3B), we generated the *bik1 aba2-1* double mutant and found that it has severe growth defects (Fig. 7H). However, we cannot get enough seeds to evaluate SnRK2 activation in the *bik1 aba2-1* double mutant. Interestingly, the *snrk2.2/3/6* triple mutant has similar growth defects compared to the *bik1* mutant under osmotic stress (Fig. 7I). These results demonstrated that BIK1 is a

central regulator of osmotic stress responses and controls multiple osmotic stress responses.

## Discussion

ABA and osmotic stress signaling both activate kinases in subgroup III of the SnRK2 family, which are core positive regulators of the osmotic stress response in plants that are inhibited by PP2Cs under unstressed conditions (Boudsocq et al, 2004; Fujii et al, 2011; Soon et al, 2012; Umezawa et al, 2009). Activation of SnRK2s requires a two-step process: release of SnRK2s from PP2C inhibition and their subsequent activation (Fig. 7J). The SnRK2-PP2C interactions occur on three major interfaces (Fig. EV5E). The first interface is between the SnRK2 activation loop and the PP2C catalytic cleft; the second interface is created by the SnRK2 residues near the catalytic cleft and the tryptophan lock of the PP2Cs; and the third interface is between the SnRK2 αG helix and the PP2C loop region adjacent to the tryptophan lock (Soon et al, 2012). Therefore, the release of SnRK2 proteins from PP2C inhibition requires disruption of the three SnRK2-PP2C interfaces.

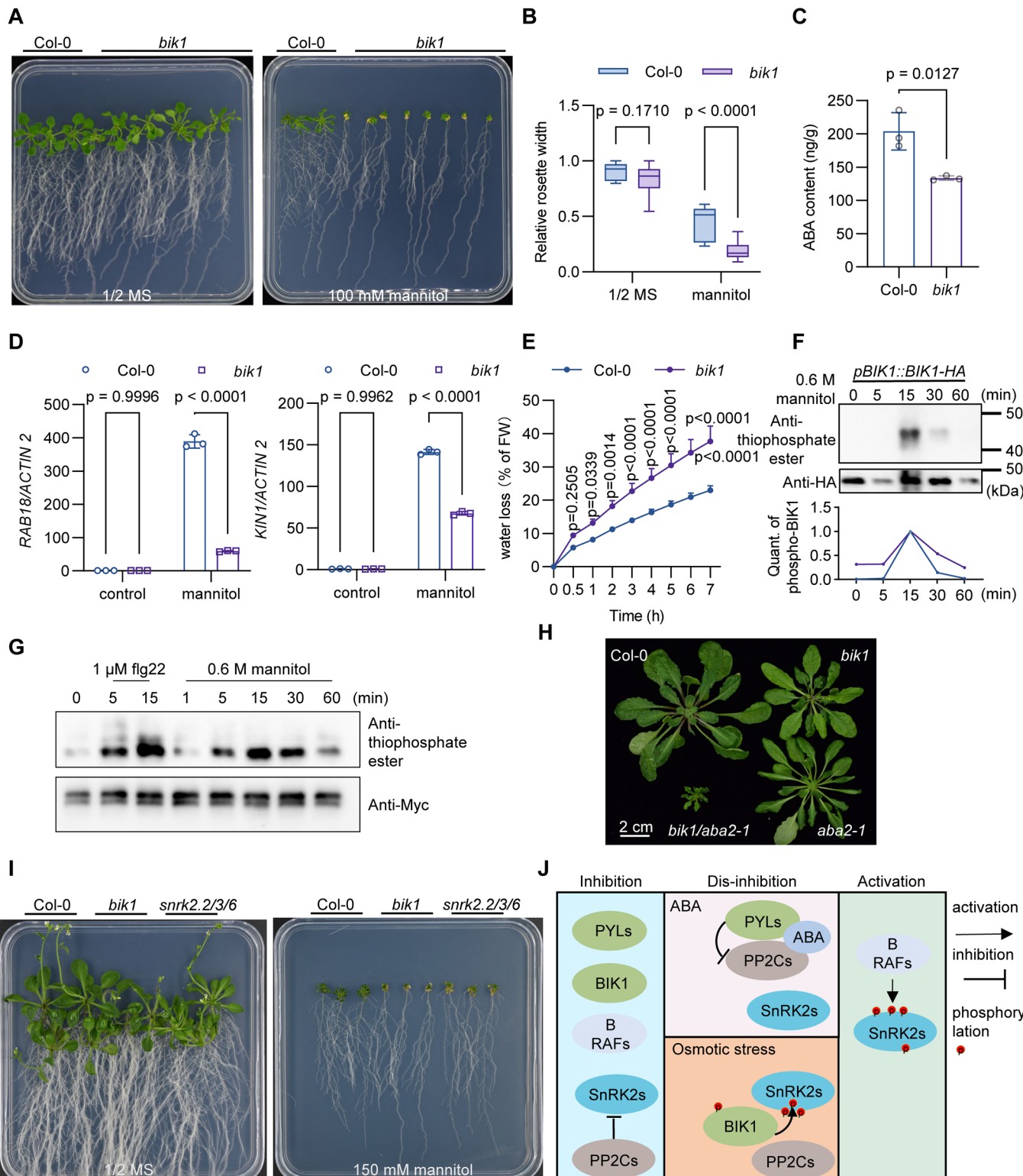

Our results show that the receptor-like cytoplasmic kinase BIK1 releases PP2C-mediated inhibition of SnRK2.6 via phosphorylation regulation, which may circumvent the PP2C-mediated inhibition of SnRK2s. BIK1 phosphorylates SnRK2.6 at Y163 and Y182; the Y182 residue is spatially close to the tryptophan lock of the PP2Cs, which is critical for its interaction with SnRK2s (Figs. 4 and 5). Phosphorylation of Y182 could increase the hydrophilicity and negative charge in the area and potentially repel the docking of the nonpolar tryptophan lock (Fig. 4G), which disrupts the second interface of PP2C-SnRK2. In contrast, the ABA-mediated SnRK2 release is exerted by ABA

Figure 7. **BIK1 controls osmotic stress responses.**

(A, B) Plant growth of Col-0 (WT) and *bik1* mutant seedlings, 15 days after the seedlings were transferred from ½ MS medium to ½ MS medium with or without 100 mM mannitol (A). The relative rosette widths were quantified (B). Data are represented by box-and-whisker plots where boxes extend from the 25th to 75th percentiles and bars within the boxes represent the medians; the whiskers represent the minimum and the maximal range. Values are means ± SD ($n ≥ 12$ seedlings). Two-way ANOVA followed by Tukey's test. (C) ABA levels in WT and *bik1* mutant seedlings after a 12-h treatment with 300 mM mannitol. Values are means ± SD ($n = 3$ independent biological repeats). Two-tailed unpaired *t* tests. (D) Relative expression of stress-responsive genes in WT and *bik1* mutant seedlings after 6 h of 300 mM mannitol treatment. Values are means ± SD ($n = 3$ independent biological repeats). Two-way ANOVA followed by Tukey's test. (E) Cumulative transpirational water loss from detached rosettes of 5-week-old WT and *bik1* mutant plants. Error bars, SD ($n = 3$ independent biological repeats). The measure of the center for the error bars is a mean of three repeats. Two-way ANOVA followed by Tukey's test. (F, G) Phosphorylation activity of HA-tagged or Myc-tagged BIK1 immunoprecipitated from 9-day-old *proBIK1:BIK1-HA* or *35S:BIK1-Myc* transgenic plants treated with 0.6 M mannitol for the indicated times. After in vitro kinase assays with ATPγS and the following PNBM alkylation, the thiophosphate ester groups on the substrate were detected by anti-thiophosphate ester antibody. The anti-HA and anti-Myc antibodies were used as loading controls. Quantification of phospho-BIK1 represents the ratio of band intensity of phosphorylated BIK1 to that of anti-HA, normalized by the value at 15 min. Plots represent two independent experiments. (H) Morphology of 5-week-old soil-grown WT, *aba2-1* and *bik1* mutants, and the *bik1/aba2-1* double mutant under short-day conditions. Scale bars, 2 cm. (I) The *snrk2.2/3/6* triple mutant has a similar growth phenotype as the *bik1* mutant under osmotic stress. Plant growth of WT, *snrk2.2/3/6* and *bik1* mutant seedlings, 20 days after the seedlings were transferred from ½ MS medium to ½ MS medium with or without 150 mM mannitol. (J) A model illustrating the release of SnRK2s from PP2C inhibition by either BIK1 under osmotic stress or PYLs under ABA treatment. Under unstressed conditions, clade A PP2Cs bind to and inhibit SnRK2s (left panel). In the presence of ABA, ABA receptors in the PYL family bind to and inhibit the PP2Cs, and in turn trigger the release of SnRK2s (upper middle panel). Under osmotic stress, the activated BIK1 phosphorylates SnRK2s and leads to the releasement of SnRK2s (lower middle panel). After being released from PP2C inhibition, SnRK2s are phosphorylated and activated by RAF proteins in subgroup B and other protein kinases (right panel). All experiments were repeated at least three times with similar results. Source data are available online for this figure.

receptors in the PYL family (Ma et al, 2009; Park et al, 2009), which interact with PP2Cs via interfaces similar to those of the SnRK2-PP2C complexes and in turn disrupts the first and second interfaces of SnRK2-PP2C complexes (Melcher et al, 2009; Miyazono et al, 2009; Soon et al, 2012; Yin et al, 2009). Consistent with our model, mutations in the PP2C catalytic domain at the first interface of SnRK2-PP2C (e.g., G180D in ABI1, G168D in ABI2, and G246D in HAB1) disrupt the ABA-mediated PYL-PP2C interactions without disrupting the PP2C-SnRK2 interactions (Leung et al, 1997; Ma et al, 2009; Park et al, 2009; Sheen, 1998; Umezawa et al, 2009); however, the abi1-1 mutation (ABI1$^{G180D}$) cannot avoid the BIK1-mediated disruption of the second interface of SnRK2-PP2C (Fig. 3). Although the core components for SnRK2 release (PYL versus BIK1) and their targeting proteins (PP2Cs versus SnRK2s) and targeting interfaces (first and second interfaces versus second interface) differ between ABA signaling and osmotic signaling, the ABA-sensing tryptophan lock at the second interface of PP2C-SnRK2 complexes constantly plays a central role during the inhibition and release of subgroup III SnRK2s (Melcher et al, 2009; Miyazono et al, 2009; Soon et al, 2012; Yin et al, 2009) (Figs. 4 and 5). The tryptophan lock is conserved in PP2Cs with the exception of AHG1, which is not inhibited by PYL but is responsive to DELAY OF GERMINATION1 (DOG1)-mediated repression during seed dormancy (Nishimura et al, 2018).

BIK1 is specific to SnRK2s in subgroup III, since BIK1 can hardly phosphorylate members of the subgroup I SnRK2s (which evolved later) and the *bik1* mutant phenotype is not as strong as the previously reported high-order *raf* and *snrk2* mutants (Fujii et al, 2011; Lin et al, 2021; Lin et al, 2020; Soma et al, 2020) (Fig. EV3B). This more limited phenotype suggests that additional components and mechanisms contribute to SnRK2 release under osmotic stress. Besides BIK1, PBLs and CPKs were also putative SnRK2-interacting proteins under osmotic stress (Fig. 1B); however, whether and how PBLs and CPKs regulate SnRK2 release and activation requires further study. Moreover, the plant NLR immune receptor Sw-5b mimics PYLs via its nucleotide-binding (NB) domain and interacts with NbPP2C4 to release NbSnRK2s (Huang et al, 2024). In contrast, photoactivated phyA interacts with PP2Cs to repress PYL-mediated SnRK2 release (Li et al, 2023). The ABA-independent

release of SnRK2 (e.g., BIK1-mediated release) occurs earlier than ABA accumulation (Figs. 1E and 7C,E) (Chen et al, 2020), and may function redundantly with the PYL-mediated SnRK2 release in regulating plant growth and stress responses (Fig. 7H,J). Together with previous findings, we outline a two-step activation mechanism of subgroup III SnRK2s and uncover a molecular mechanism for releasing SnRK2.6 from PP2C-mediated inhibition under osmotic stress. Further studies are needed to elucidate additional mechanisms of SnRK2 release, especially for the subgroup I SnRK2s.

After its release from inhibition, SnRK2.6 can then be activated by the B2 and B3 subgroups of RAF kinases via phosphorylation of S171 and S175 (Lin et al, 2021; Lin et al, 2020; Saruhashi et al, 2015; Takahashi et al, 2020). The B2 RAFs are constantly active and control subgroup III SnRK2 activation by ABA or mild osmotic stress; while B3 RAFs are activated under severe osmotic stress and regulate subgroup III SnRK2 activation in an ABA-independent manner (Lin et al, 2021; Soma et al, 2023; Takahashi et al, 2020). The B4 RAFs are also activated under severe osmotic stress and regulate subgroup I SnRK2 activation in an ABA-independent manner (Lin et al, 2020; Soma et al, 2020). However, the higher order mutants of B2 and B3 *RAFs* are still responsive to higher concentrations of ABA, suggesting additional protein kinases also participate in the SnRK2 activation process, e.g., BRASSINOSTEROID-INSENSITIVE 2 (BIN2) also phosphorylates and activates SnRK2.2 and 2.3 (Cai et al, 2014). Further studies are needed to elucidate additional protein kinases involved in SnRK2 activation.

In a comparison of osmotic stress signaling and ABA signaling, we may deduce that the releasement process of SnRK2 is determined by osmotic stress sensing since B2 RAFs are constitutively active and sufficient for subgroup III SnRK2 activation. Although the sensing and early signaling of osmotic stress is still unclear, recent findings suggest that it occurs at the plasma membrane. The plasma membrane-localized proteins, including the mechanically activated ion channel OSCA1 (Yuan et al, 2014), the $Ca^{2+}$ responsive phospholipid-binding BON proteins (Chen et al, 2020), ROP6 GTPase and NADPH oxidases (Martiniere et al, 2019; Smokvarska et al, 2020), and several subgroups of receptor-like kinases including BR-INSENSITIVE 1 (BRI1) ASSOCIATED RECEPTOR KINASE 1 (BAK1) (Bacete et al,

2022; Fabregas et al, 2018; Grison et al, 2019; Pei et al, 2022; Smokvarska et al, 2023; Takahashi et al, 2018; Zhao et al, 2023), participate in osmotic or drought stress signaling, although some of them may function as negative regulators (Bacete et al, 2022; Chen et al, 2016; Yu et al, 2012). Among these RLKs, BAK1 directly phosphorylates and activates BIK1 during immunity signaling (Lin et al, 2014). In addition, the mitogen-activated protein kinase kinase kinase kinase (MAP4K) SIK1 (the yeast STE20 homolog in plants) (Xiong et al, 2016) phosphorylates BIK1 and functions together with BIK1 in the activation of NADPH oxidases (Zhang et al, 2018). Interestingly, the small GTPase Cdc42 binds and activates STE20 to activate osmotic stress signaling in yeast (Raitt et al, 2000) and in Y3H assays BIK1 dramatically repressed SnRK2.6-ABI1 interaction when grown on media with 1 M mannitol (Fig. 2A), suggesting putative activation of BIK1 by STE20 or other protein kinases in yeast. BIK1 and PBLs belong to the RLCK VII subfamily. Besides SnRK2s and PP2Cs (Fig. 4D,E), RLCK VIIs are also evolutionarily conserved kinases that originated in green algae (Ngou et al, 2024), and *MpPBLa* from *Marchantia polymorpha* can complement the *Arabidopsis bik1pbl1* mutant defect in chitin-induced ROS production (Chu et al, 2023). Future studies should investigate PBL function in SnRK2 release in green algae and early land plants. Therefore, BIK1 may respond to upstream osmotic signaling components and transduce the cell surface signals to the intracellular osmotic stress responses. Further studies are required to clarify these signaling mechanisms.

BIK1 has a central role in plant stomatal responses through phosphorylation of multiple substrates. For example, BIK1 controls $Ca^{2+}$ signals via two types of calcium channels, CNGC2/4 and OSCA1.3 (Thor et al, 2020; Tian et al, 2019); BIK1 also mediates ROS production via phosphorylation of RBOHD (Kadota et al, 2014), which in turn inhibits ABI2 phosphatase activity (Bi et al, 2022). Since SnRK2.6 is also involved in stomatal immunity, the BIK1-mediated SnRK2 activation may also contribute to the stomatal response during plant immunity. Future studies are needed to clarify whether and how BIK1-mediated SnRK2 phosphorylation controls stomatal immunity.

To summarize, here we illustrated a molecular mechanism for releasing SnRK2.6 from PP2C-mediated inhibition under osmotic stress. The plasma membrane-localized receptor-like kinase BIK1 directly phosphorylates SnRK2.6 at tyrosine residues adjacent to where a critical tryptophan of PP2Cs that locks in SnRK2-PP2C interactions. This means that BIK1 functions as a central regulator of osmotic stress responses and links cell surface signaling to internal responses.

# Methods

### Reagents and tools table

| Reagent or resource | Source | Identifier |
| --- | --- | --- |
| **Antibodies** | | |
| ACTIN (N) antibody, rabbit polyclonal | Abiocode | Cat#: R3772-1P |
| Tubulin A (N) antibody, mouse monoclonal | Abiocode | Cat#: M0267-1a |
| GST Tag monoclonal antibody | Thermo Fisher | Cat#: 13-6700 |
| Anti-phospho-S175-SnRK2.6 antibody | ABclonal | Antigen: SVLHSQPK-pSTVGTP |

| Reagent or resource | Source | Identifier |
| --- | --- | --- |
| Anti-thiophosphate ester antibody | Abcam | Cat#: ab92570 |
| Anti-HA nanobody magarose beads | ShareBio | Cat#: SB-NM003 |
| Alpaca anti-GFP VHH magnetic beads | ShareBio | Cat#: SB-NM001 |
| Anti-MYC nanobody magarose beads | ShareBio | Cat#: SB-NM019 |
| Anti-HA nanobody magarose beads | ShareBio | Cat#: SB-NM003 |
| Anti-Myc-HRP | GNI | Cat#: GNI4310-MC |
| Anti-HA-peroxidase | Roche | Cat#: 12013819001 |
| Anti-SnRK2,2, SnRK2,3, SnRK2,6 | Agrisera | Cat#: AS14 2783 |
| GAL4 DNA Binding Domain Polyclonal Antibody | Abbkine | Cat#: ABP57232 |
| Anti-Luciferase antibody produced in rabbit | Sigma-Aldrich | Cat#: L0159 |
| Living Colors® A.v. Monoclonal Antibody (JL-8) | Clontech | Cat#: 632381 |
| Phosphotyrosine Monoclonal Antibody (pY20), HRP | Thermo Fisher Scientific | Cat#: 03-7720 |
| Goat anti-mouse IgG (H + L)-HRP conjugate | Bio-Rad | Cat#: 172-1011 |
| Goat anti-rabbit IgG (H + L)-HRP conjugate | Bio-Rad | Cat#: 172-1019 |
| **Bacterial and virus strains** | | |
| *Escherichia coli* BL21 | Widely distributed | N/A |
| *Agrobacterium tumefaciens* (strain GV3101) | Widely distributed | N/A |
| *Escherichia coli* DH5a | Widely distributed | N/A |
| **Chemicals, peptides, and recombinant proteins** | | |
| Murashige & Skoog (MS) basal salt mixture | Phytotechlab | Cat#: M524 |
| D-Mannitol | Sinopharm Chemical Reagent | Cat#: 63008818 |
| NaCl | Sinopharm Chemical Reagent | Cat#: 10019308 |
| Sucrose | Sinopharm Chemical Reagent | Cat#: 10021418 |
| D-Sorbitol | Sangon Biotech | Cat#: A610491 |
| Yeast nitrogen base without amino acids | Sigma-Aldrich | Cat#: Y0626 |
| DO Supplement –His/–Leu/–Trp | Clontech | Cat#: 630419 |
| DO Supplement –Leu/–Trp | Clontech | Cat#: 630417 |
| DO Supplement -Ade/-His/-Leu/-Met/-Trp | Coolaber | Cat#: PM2350 |
| Sodium fluoride | Sigma-Aldrich | Cat#: S7920 |
| Abscisic acid | Sigma-Aldrich | Cat#: A1049 |
| Leupeptin | Sigma-Aldrich | Cat#: 62070 |

| Reagent or resource | Source | Identifier |
|---|---|---|
| Antipain | Sigma-Aldrich | Cat#: 10791 |
| Aprotinin | Sigma-Aldrich | Cat#: 10981532001 |
| ATP-gamma-S, kinase substrate | Abcam | Cat#: 138911 |
| p-Nitrobenzyl mesylate, alkylation reagent | Abcam | Cat#: 138910 |
| ATP, [γ-32P]- 6000 Ci/mmol | PerkinElmer | Cat#: NEG502Z500UC |
| Trizol reagent | Invitrogen | Cat#: 15596018 |
| Firefly D-luciferin | NanoLight | CAS#: 2591-17-5 |
| **Critical commercial assays** | | |
| Hifair™ III 1st Strand cDNA Synthesis SuperMix for qPCR (gDNA Digester Plus) | Yeasen | Cat#: 11141ES60 |
| pEASY-Basic Seamless Cloning and Assembly Kit | TransGen Biotech | Cat#: CU201-02 |
| PrimeSTAR Max DNA Polymerase | Takara Bio | Cat#: R045A |
| 2X Universal SYBR Green Fast qPCR Mix | Abclonal | Cat#: RK21203 |
| LumiQ HRP substrate solution kit | ShareBio | Cat#: SB-WB012 |
| **Experimental models: organisms/strains** | | |
| *Arabidopsis thaliana*: Col-0 | Widely distributed | N/A |
| *Arabidopsis thaliana*: aba2-1 | Gonzalez-Guzman et al, 2002 | Substitution G1464A. |
| *Arabidopsis thaliana*: snrk2.6 | Fujii et al, 2011 | SALK_008068 |
| *Arabidopsis thaliana*: snrk2.2/3/6 | Fujii et al, 2011 | N/A |
| *Arabidopsis thaliana*: bik1 | Veronese et al, 2006 | N/A |
| *Arabidopsis thaliana*: bik1/aba2-1 | This study | N/A |
| *Arabidopsis thaliana*: abi1-1 | Luo et al, 2014 | Substitution G539A. |
| *Arabidopsis thaliana*: ost1-3/SnRK2.6-GFP | Zhao et al, 2018a | N/A |
| *Arabidopsis thaliana*: pBIK1:BIK1-HA | Yu et al, 2022 | N/A |
| *Arabidopsis thaliana*: pSTOP1:STOP1-HA | Zhang et al, 2019 | N/A |
| *Arabidopsis thaliana*: Super:OST1-Myc | Ding et al, 2015 | N/A |
| *Arabidopsis thaliana*: SnRK2.6pro: SnRK2.6/snrk2.6 | This study | N/A |
| *Arabidopsis thaliana*: SnRK2.6pro: SnRK2.6$^{Y163F}$/snrk2.6 | This study | N/A |
| *Arabidopsis thaliana*: SnRK2.6pro: SnRK2.6$^{Y163D}$/snrk2.6 | This study | N/A |
| *Arabidopsis thaliana*: SnRK2.6pro: SnRK2.6$^{Y182F}$/snrk2.6 | This study | N/A |
| *Arabidopsis thaliana*: SnRK2.6pro: SnRK2.6$^{Y182D}$/snrk2.6 | This study | N/A |
| *Arabidopsis thaliana*: SnRK2.6pro: SnRK2.6$^{Y163/182F}$/snrk2.6 | This study | N/A |
| *Arabidopsis thaliana*: SnRK2.6pro: SnRK2.6$^{Y163/182D}$/snrk2.6 | This study | N/A |
| *Arabidopsis thaliana*: SnRK2.6pro: SnRK2.6$^{Y163F/Y182D}$/snrk2.6 | This study | N/A |
| *Arabidopsis thaliana*: SnRK2.6pro: SnRK2.6$^{Y163D/Y182F}$/snrk2.6 | This study | N/A |
| *Arabidopsis thaliana*: 35Spro: BIK1-Myc/Col-0 | This study | N/A |
| **Oligonucleotides** | | |
| Primers used in this study | This study, Dataset EV3 | N/A |
| **Recombinant DNA** | | |
| *pCAMBIA1305-SnRK2.6pro: SnRK2.6* | This study | N/A |
| *pCAMBIA1305-SnRK2.6pro: SnRK2.6$^{Y163F}$* | This study | N/A |
| *pCAMBIA1305-SnRK2.6pro: SnRK2.6$^{Y163D}$* | This study | N/A |
| *pCAMBIA1305-SnRK2.6pro: SnRK2.6$^{Y182F}$* | This study | N/A |
| *pCAMBIA1305-SnRK2.6pro: SnRK2.6$^{Y182D}$* | This study | N/A |
| *pCAMBIA1305-SnRK2.6pro: SnRK2.6$^{Y163/182F}$* | This study | N/A |
| *pCAMBIA1305-SnRK2.6pro: SnRK2.6$^{Y163/182D}$* | This study | N/A |
| *pCAMBIA1305-SnRK2.6pro: SnRK2.6$^{Y163F/182D}$* | This study | N/A |
| *pCAMBIA1305-SnRK2.6pro: SnRK2.6$^{Y163D/Y182F}$* | This study | N/A |
| *pCAMBIA1300-35Spro: BIK1-Myc* | This study | N/A |
| *pCAMBIA1300-35S-SnRK2.6-YFP$^n$* | This study | N/A |
| *pCAMBIA1300-35S-BIK1-YFP$^c$* | This study | N/A |
| *pCAMBIA1300-35S-FER-YFP$^c$* | Chen et al, 2022 | N/A |
| *pGEX-6P-1- SnRK2.2$^{K52N}$* | This study | N/A |
| *pGEX-6P-1- SnRK2.3$^{K51N}$* | This study | N/A |
| *pGEX-6P-1-SnRK2.6* | This study | N/A |
| *pGEX-6P-1-SnRK2.6$^{K50N}$* | This study | N/A |
| *pGEX-6P-1-SnRK2.4$^{K33N}$* | This study | N/A |
| *pGEX-6P-1- SnRK2.6$^{K50N/Y163F}$* | This study | N/A |
| *pGEX-6P-1- SnRK2.6$^{K50N/Y82F}$* | This study | N/A |
| *pGEX-6P-1- SnRK2.6$^{K50N/Y163/182F}$* | This study | N/A |
| *pGEX-6P-1- SnRK2.6$^{Y163F}$* | This study | N/A |
| *pGEX-6P-1- SnRK2.6$^{Y182F}$* | This study | N/A |
| *pGEX-6P-1- SnRK2.6$^{Y163/182F}$* | This study | N/A |
| *pGEX-6P-1- SnRK2.6$^{Y163D}$* | This study | N/A |
| *pGEX-6P-1- SnRK2.6$^{Y182D}$* | This study | N/A |
| *pGEX-6P-1- SnRK2.6$^{Y163/182D}$* | This study | N/A |
| *pGEX-6P-1- SnRK2.6$^{Y163E}$* | This study | N/A |
| *pGEX-6P-1- SnRK2.6$^{Y182E}$* | This study | N/A |
| *pGEX-6P-1- SnRK2.6$^{Y163/182E}$* | This study | N/A |
| *pGEX-6P-1- ABI1* | This study | N/A |
| *pGEX-6P-1- ABI1$^{W300A}$* | This study | N/A |
| *pGEX-6P-1- PP2CA* | This study | N/A |
| *pGEX-6P-1- PP2CA$^{W280A}$* | This study | N/A |

| Reagent or resource | Source | Identifier |
|---|---|---|
| *pGEX-6P-1- ABF2*[fragment] | Zhao et al, 2018a | N/A |
| *pET-32a- BIK1* | This study | N/A |
| *pMAL-C2X-BIK1* | This study | N/A |
| *pMAL-C2X-SnRK2.6* | This study | N/A |
| *pMAL-C2X-SnRK2.6*[Y163F] | This study | N/A |
| *pMAL-C2X-SnRK2.6*[Y182F] | This study | N/A |
| *pMAL-C2X-SnRK2.6*[Y163/182F] | This study | N/A |
| *pCAMBIA1305-GUS-nLUC* | This study | N/A |
| *pCAMBIA1305-SnRK2.6-nLUC* | This study | N/A |
| *pCAMBIA1305-SnRK2.6*[13 A/Fs]*-nLUC* | This study | N/A |
| *pCAMBIA1305-SnRK2.6*[13Ds]*-nLUC* | This study | N/A |
| *pCAMBIA1305-BIK1-nLUC* | This study | N/A |
| *pCAMBIA1305-cLUC-FER ECD* | This study | N/A |
| *pCAMBIA1305-cLUC-ABI1* | This study | N/A |
| *pCAMBIA1305-cLUC-abi1-1* | This study | N/A |
| *pCAMBIA1305-cLUC-HAB1* | This study | N/A |
| *pCAMBIA1305-35Spro:BIK1-GFP* | This study | N/A |
| *pCAMBIA1305-35Spro:BIK1*[K105/106R]*-GFP* | This study | N/A |
| *pGADT7-ABI1* | This study | N/A |
| *pGADT7-ABI1*[W300A] | This study | N/A |
| *pGADT7-ABI1*[W300R] | This study | N/A |
| *pGADT7-HAB1* | This study | N/A |
| *pGADT7-HAB1*[W385A] | This study | N/A |
| *pGADT7-abi1-1* | This study | N/A |
| *pGBKT7-SnRK2.6* | This study | N/A |
| *pGBKT7-SnRK2.6*[13A/Fs] | This study | N/A |
| *pGBKT7-SnRK2.6*[13Ds] | This study | N/A |
| *pGBKT7-BIK1* | This study | N/A |
| *pBridge-BD-SnRK2.6* | This study | N/A |
| *pBridge-BD-SnRK2.6-pMET-BIK1* | This study | N/A |
| *pBridge-BD-SnRK2.6-pMET-BIK1*[K105/106R] | This study | N/A |
| **Software and algorithms** | | |
| ImageJ | Schneider et al, 2012 | RRID: SCR_003070 |
| GraphPad Prism 8 | Informer Technologies, Inc. | RRID: SCR_002798 |
| TBtools | Chen et al, 2023 | N/A |

## Plant materials and growth conditions

All *Arabidopsis* plants are in the Col-0 ecotype background. The T-DNA insertion mutants *bik1*, *ost1-3/snrk2.6*, *snrk2.2/3/6*, and the point-mutation mutant *aba2-1* have been reported (Fujii et al, 2011; González-Guzmán et al, 2002; Veronese et al, 2006). The gain of function mutant, *abi1-1*, has been generated previously (Luo et al, 2014). The *bik1* was crossed with *aba2-1* to generate double

mutant. The transgenic lines, including *ost1-3/SnRK2.6-GFP*, *pBIK1:BIK1-HA*, *pSTOP1:STOP1-HA*, and *Super:OST1-Myc* plants have also been described previously (Ding et al, 2015; Yu et al, 2022; Zhang et al, 2019; Zhao et al, 2018a). For *ost1-3/snrk2.6* complementation, a 4166-bp genomic DNA fragment containing 1461 bp upstream of the translation start site and 411 bp downstream of the stop codon of *SnRK2.6* was cloned into the *Kpn*I and *Eco*RI sites of *pCAMBIA1305* using indicated primers (Dataset EV3). In this study, we generated transgenic lines with native promoter-driven wild-type and mutated *SnRK2.6* in the *ost1-3* mutant background, including: *SnRK2.6*, *SnRK2.6*[Y163F], *SnRK2.6*[Y182F], *SnRK2.6*[Y163F/Y182F], *SnRK2.6*[Y163D], *SnRK2.6*[Y182D], *SnRK2.6*[Y163D/Y182D], *SnRK2.6*[Y163F/Y182D], and *SnRK2.6*[Y163D/Y182F] transgenic plants. In addition, full-length *BIK1* was amplified and fused with *Myc* with the *pCAMBIA1305* vector to generate *35S:BIK1-Myc*. The constructed vectors were transformed into Arabidopsis plants by floral dip (Clough and Bent, 1998). The *35S:BIK1-Myc* construct was transformed to the wild-type Col-0.

## Plant growth conditions and treatments

Seeds were sterilized with 6% sodium hypochlorite for 10 min. After rinsed three times in sterile-deionized water, seeds were grown horizontally on 0.75% (w/v) agar or vertically on 1.2% (w/v) agar containing half-strength Murashige and Skoog (1/2 MS) (PhytoTech), 1% sucrose, pH5.7, and stratified at 4 °C for 3 days. Three- or four-day-old vertical-grown seedlings were transferred from 1/2 MS medium to medium with or without 100 mM or 150 mM mannitol for growth phenotype analyses. The seedlings were grown on medium in a Percival CU36L5 incubator at 23 °C under long-day conditions (16-h light/8-h dark). Plants were also grown in soil in a growth room at 23 °C with a 65–80% relative humidity under long-day conditions. For the detached leaf water loss assay, plants were grown in soil in a growth room at 23 °C with a 65–80% relative humidity under short-day conditions (8-h light/16-h dark).

*Nicotiana benthamiana* was grown in soil in a growth room under a 16-h light/8-h dark photoperiod at 23 °C.

## Yeast two-hybrid (Y2H) assay

The coding sequences of *BIK1*, *SnRK2.6*, *SnRK2.6*[13A/Fs], and *SnRK2.6*[13Ds], were amplified by PCR and cloned into the *pGBKT7* vector; while the coding sequences of *ABI1*, *ABI1*[W300A], *ABI1*[W300R], *HAB1*, and *HAB1*[W385A], were amplified by PCR and cloned into the *pGADT7* vector, using indicated primers (Dataset EV3). According to the manufacturer's instructions, the Y2H assays were performed using the Yeastmaker Yeast Transformation System 2 (Clonetech). Briefly, the *pGBKT7-SnRK2.6*, *pGBKT7-SnRK2.6*[13A/Fs], *pGBKT7-SnRK2.6*[13Ds] or *pGBKT7-BIK1* constructs were introduced into the yeast strain Gold expressing the *pGADT7-ABI1*, *pGADT7-ABI1*[W300A], *pGADT7-ABI1*[W300R], *pGADT7-HAB1*, *pGADT7-HAB1*[W385A] constructs using the polyethylene glycol/LiAc-mediated yeast transformation. The yeast colonies containing both indicated genes were selected on the synthetic defined (SD) medium without leucine and tryptophan (SD-LW), and the interactions were analyzed on the SD medium without leucine, tryptophan, and histidine (SD-LWH), or without adenine, leucine, tryptophan, and histidine (SD-ALWH).

## Yeast three-hybrid (Y3H) assay

The coding sequences of *SnRK2.6*, and wild-type or *K105R/K106R* mutated *BIK1*, were amplified by PCR and cloned into MCS I and MCS II of the pBridge vector using indicated primers (Dataset EV3), respectively. The full-length *ABI1* and *ABI1^{G180D}* (*abi1-1*) were cloned into a *pGADT7* vector using indicated primers (Dataset EV3). The *pGADT7* constructs were co-transformed with the *pBridge* constructs into the yeast strain AH109 in indicated combinations and screened on a medium without leucine, tryptophan, and methionine (SD-LWM). The interactions were tested on the SD medium without adenine, leucine, tryptophan, histidine, and methionine (SD-ALWHM), without or with 1 M mannitol.

## Protein extraction and expression analyses in yeast cells

Protein extraction experiments were performed according to the previous method (Zhang et al, 2011). Yeast cells were collected at $OD_{600} = 1.0$ and then treated with 2.0 M LiAC solution for 1 min on ice, then replaced with 0.4 M NaOH for 5 min on ice. Subsequently, cells were resuspended in a 2×SDS-PAGE loading buffer and boiled for 10 min. For the relative level of protein expression, *pGBKT7-SnRK2.6* was immunoblotted with an anti-GAL4 DNA Binding Domain polyclonal antibody (anti-BD, ABP57232, Abbkine, 1:5000) in Y3H assay or anti-Myc-HRP antibody (GNI4310-MC, GNI, 1:3000) in Y2H assay, and AD-HA-ABI1 or HA-BIK1 were detected with an anti-HA-Peroxidase (12013819001, Roche, 1:5000) by western blot. The protein level of tubulin was detected by anti-tubulin (M0267-1a, Abiocode, 1:5000), and was used as a loading control.

## Co-immunoprecipitation (IP) assay

The total proteins were extracted from *pBIK1:BIK1-HA* transgenic plants using 2×IP buffer (40 mM Tris-HCl, pH 7.5, 200 mM NaCl, 2 mM EDTA, 2 mM EGTA, 2 mM $Na_3VO_4$, 2 mM NaF, 20 mM β-glycerophosphate, 4 mg/mL leupeptin, 4 mg/mL antipain, 4 mg/mL aprotinin, 0.2% Tween 20, and 1×InStab™ Protease Cocktail) and incubated with anti-HA Nanobody Magarose Beads (SB-NM003, ShareBio, Shanghai) for 2 h. The beads were washed three times with 1×IP buffer. The immunoprecipitated proteins were analyzed by SDS-PAGE and detected with anti-SnRK2.2/3/6 (AS14 2783, Agrisera, 1:5000), anti-HA-Peroxidase (12013819001, Roche, 1:5000), and anti-actin (R3772-1P, Abiocode, 1:10,000) antibodies.

## Split luciferase (LUC) complementation assay

The coding sequences of *GUS*, *SnRK2.6*, *SnRK2.6^{13A/Fs}*, *SnRK2.6^{13Ds}*, and *BIK1*, were amplified by PCR and cloned into the *pCAMBIA-35S-nLUC* vector; while the coding sequences of *FER ECD*, *BIK1*, *ABI1*, *abi1-1*, *ABI1^{W300A}*, *ABI1^{W300R}*, and *HAB1*, were amplified by PCR and cloned into the *pCAMBIA-35S-cLUC* vector, using indicated primers (Dataset EV3). Split LUC complementation assay was performed by transient expression of indicated combinations of constructs in tobacco leaves through *Agrobacterium*-mediated infiltration as previously described (Zhao et al, 2018a). Two days after infiltration, luciferase activity was detected with a CCD camera by applying firefly D-luciferin (NanoLight).

The *cLUC-FER ECD* and *GUS-nLUC* were used as negative controls.

To detect protein expression levels, total proteins were extracted from *N. benthamiana* leaves, and separated by SDS-PAGE, followed by immunoblotting analyses with an anti-LUC antibody (L0159, Sigma-Aldrich, 1;5000), anti-GFP antibody (JL-8, Clontech, 1:5000), and anti-actin antibody (R3772-1P, Abiocode, 1:10,000).

## Immunoprecipitation-mass spectrometry (IP-MS)

Nine-day-old *ost1-3*/*SnRK2.6-GFP* or *Super:OST1-Myc* transgenic seedlings were treated with 50 µM ABA or 600 mM mannitol or left untreated. Total proteins were extracted using a 2×IP buffer. The SnRK2.6-GFP and OST1-Myc proteins were immunoprecipitated using Alpaca anti-GFP VHH beads (SB-NM011, ShareBio, China) or anti-Myc beads (SB-NM019, ShareBio, China) and were used for high-performance liquid chromatography–MS analysis (timsTOF Pro2, Bruker).

## Bimolecular fluorescence complementation (BiFC)

For the BiFC assay, the full-length coding sequences of *SnRK2.6* and *BIK1* were cloned into *pCAMBIA1300s*-YC and *pCAMBIA1300s*-YN using indicated primers (Dataset EV3), respectively. The resultant cYFP-BIK1 and nYFP-SnRK2.6 constructs were coexpressed in *N. benthamiana* leaves via *Agrobacterium*-mediated infiltration. In addition, equal-volume suspensions of *Agrobacterium* strain GV3101 harboring a plasma member marker (*PIP2a-mRFP*) were mixed for the *Agrobacterium*-mediated infiltration. The plants were grown in the growth room for 48 h after infiltration, and BiFC fluorescence signals were then observed using a ZEISS LSM 880.

## In vitro phosphorylation assay

For the in vitro phosphorylation assays, recombinant protein kinases, including MBP-tagged BIK1, His-tagged BIK1, or GST-tagged wild-type and mutated SnRK2.6, were incubated with indicated substrates, such as GST-tagged SnRK2.6^{K50N}, SnRK2.2^{K52N}, SnRK2.3^{K51N}, ABI1, or ABF2 fragment (Gly 73 to Gln 119) in 25 µl of reaction buffer [50 mM Tris-HCl, pH 7.5, 20 mM $MgCl_2$, 0.25 mM DTT, 1 µM ATP, 5 mCi (γ-$^{32}$P) ATP] at 30 °C. After incubation for 2 h, the proteins were separated by SDS-PAGE. After electrophoresis, the gel was dried for 1 h on filter paper and then exposed to a phosphor-imager overnight. Radioactivity was detected with a Personal Molecular Imager (Bio-Rad, Hercules, CA) or Typhoon biomolecular imager (GE). For in vitro phosphorylation assays using anti phosphor-tyrosine antibody, the gel was transferred to a PVDF membrane for western blots after electrophoresis. The anti-phospho-tyrosine antibody (03-7720, Thermo Fisher Scientific, 1:3000) was used to detect phosphorylated proteins. The anti-GST antibody (13-6700, Thermo Fisher Scientific, 1:7500) was used to detect the loading of GST-tagged proteins.

For in vitro phosphorylation assays using the ATP analog ATPγS, recombinant GST-tagged SnRK2.6, SnRK2.6^{Y163F}, SnRK2.6^{Y182F}, SnRK2.6^{Y163F/Y182F}, SnRK2.6^{Y163D}, SnRK2.6^{Y182D}, or SnRK2.6^{Y163D/Y182D}, were incubated with 20 ng of GST-tagged

ABF2 fragment in 25 µl of reaction buffer [50 mM Tris-HCl, pH 7.0, 20 mM MgCl$_2$, 0.25 mM DTT, 1.5 mM ATPγS (Abcam, ab138911)] at 30 °C for 30 min. Then, PNBM (Abcam, ab138910) in DMSO was added to afford a final concentration of 2.5 mM with 5% DMSO. After incubation for 1 h, the proteins were separated by SDS-PAGE. After electrophoresis, the gel was transferred to a PVDF membrane for western blots. The anti-thiophosphate ester antibody (ab92570, Abcam, 1:10,000) was used to detect phosphorylated proteins. The anti-GST antibody (13-6700, Thermo Fisher Scientific, 1:7500) was used to detect the loading of GST-tagged proteins.

To identify putative phosphorylation sites of SnRK2.6 mediated by BIK1 in vitro, recombinant MBP-tagged BIK1 was incubated with GST-tagged SnRK2.6$^{K50N}$ in 25 µl of reaction buffer [50 mM Tris-HCl, pH 7.5, 20 mM MgCl$_2$, 0.25 mM DTT, 1 mM γ-($^{18}O_4$) ATP] at 30 °C. After incubation for 2 h, the proteins were digested with Trypsin (V5280, Promega) with a final 1:50 (wt/wt) enzyme-to-protein ratio and kept overnight. The putative phosphorylation sites of SnRK2.6$^{K50N}$ were identified by Q Exactive HF-X.

## Measurement of water loss

To measure water loss, detached rosettes of 5-week-old plants were placed in plastic weighing dishes and left on the laboratory bench with light. Fresh weight was weighed at the indicated time points. Water loss was expressed as a percentage of initial fresh weight.

## Measurement of ABA content

Nine-day-old seedlings were transferred from 1/2 MS medium to medium with 300 mM mannitol for 12 h. Tissues were ground in liquid nitrogen and then extracted using ethyl acetate. The ABA concentration was measured by AB 5500 Q-TRAP (Plant Proteomics and Metabolomics Core Facility, CEMPS, CAS).

## RNA extraction and expression analyses

Total RNA was extracted from 9-day-old seedlings using Trizol (15596018, Invitrogen). Reverse transcription reactions were performed with 1 µg of total RNA using Hifair®III 1st Strand cDNA Synthesis SuperMix (11141ES60, Yeasen, China). RT-qPCR was then performed for 40 cycles using indicated primers (Dataset EV3). The real-time qPCR assay was performed with 2×Universal SYBR Green Fast qPCR Mix (RK21203, Abclonal, China) according to the manufacturer's instructions.

## Immunoprecipitated kinase assay

The immunoprecipitated kinase assays of BIK1-HA or BIK1-Myc were performed as previously described with some modifications (Zhao et al, 2018a). Nine-day-old *pBIK1:BIK1-HA* or *35S:BIK1-Myc* seedlings were treated with or without 0.6 M mannitol for the indicated times. Samples (about 0.5 g) were collected and grounded in liquid nitrogen. Total protein was extracted in 1.5 mL 2×IP buffer. After centrifuged at 13,000 × g for 20 min, the supernatants were incubated with HA beads (SB-NM003, ShareBio, China) or Myc beads (SB-NM019, ShareBio, China) for 3 h at 4 °C. After incubation, the beads were washed with 1×IP buffer 3 times, followed by kinase buffer (25 mM Tris-HCl, pH 7.5, 10 mM MgCl$_2$,

0.25 mM DTT) one more time. Then, the anti-HA beads or anti-Myc beads were incubated in kinase buffer with the ATP analog ATPγS at room temperature for 30 min for in vitro phosphorylation assays, followed by the PNBM alkylation. The proteins were separated by SDS-PAGE. After electrophoresis, the gel was transferred to a PVDF membrane for western blots. The anti-thiophosphate ester antibody (ab92570, Abcam, 1:10,000) was used to detect the phosphorylated proteins. The protein levels of samples were detected with the anti-HA (12013819001, Roche, 1:5000) antibody or anti-Myc-HRP antibody (GNI4310-MC, GNI, 1:3000).

## Quantification and statistical analysis

Statistical parameters are reported in the figures and figure legends. Statistical analysis was performed using GraphPad Prism 9. The two-tailed Student's *t* test was used to analyze the statistical significance between the two groups. One-way or two-way ANOVA was used to analyze the statistical significance between more than two groups.

## Data availability

This study includes no data deposited in external repositories. All data reported in this paper will be shared by the lead contact upon request. This paper does not report the original code. Any additional information required to reanalyze the data reported in this paper is available from the lead contact upon request.

The source data of this paper are collected in the following database record: biostudies:S-SCDT-10_1038-S44318-024-00277-0.

## Peer review information

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

## Acknowledgements

The authors thank members of the Zhao Lab for helpful discussions. This work was supported by the Science and Technology Commission of Shanghai Municipality (Grant No. 22ZR1481400), the Strategic Priority Research Program of the Chinese Academy of Sciences (Grant No. XDB063020102), the Project of Stable Support for Youth Teams in Basic Research Field of the Chinese Academy of Sciences (YSBR-119), and the Shanghai Center for Plant Stress Biology from the Chinese Academy of Sciences.

## Author contributions

**Guo-Jun Li**: Data curation; Formal analysis; Validation; Investigation; Visualization; Writing—original draft. **Kong Chen**: Data curation; Formal analysis; Investigation; Visualization. **Shujing Sun**: Data curation; Formal analysis; Investigation. **Yang Zhao**: Conceptualization; Supervision; Funding acquisition; Writing—original draft; Project administration; Writing—review and editing.

Source data underlying figure panels in this paper may have individual authorship assigned. Where available, figure panel/source data authorship is listed in the following database record: biostudies:S-SCDT-10_1038-S44318-024-00277-0.

## Disclosure and competing interests statement

The authors declare no competing interests.

# Expanded View Figures

**Figure EV1.   BIK1 interacts with SnRK2.6 and mediates SnRK2 activation under osmotic stress.** ▶

(A) Immunoprecipitation followed by mass spectrometry was used to identify candidate OST1-Myc-interacting proteins under control or 600 mM mannitol treatment. Venn diagrams show the numbers of proteins identified from *Super:OST1-Myc* transgenic seedlings (left panel). Representative families of protein kinases were enriched among the SnRK2.6-Myc interacting proteins detected only under osmotic stress (right panel). (B) BIK1 mainly interacts with SnRK2.6 at the plasma membrane in *N. benthamiana* leaves, as shown using bimolecular fluorescence complementation (BiFC) assay. The plasma membrane marker PIP2a-mRFP was co-transformed, and cells expressing both fluorescent proteins were analyzed. Scale bars, 20 μm. (C, D) Replications of experiments showing reduced SnRK2 activation in the *bik1* mutant under osmotic stress. Phosphorylation of SnRK2s induced by 0.6 M mannitol was detected with the anti-phospho-S175-SnRK2s antibody (top), using total proteins extracted from wild-type (WT) Col-0 and *bik1* mutant seedlings. Actin was used as the loading control (bottom). (E, F) ABA-induced SnRK2 activation was not reduced in the *bik1* mutant compared with the WT. Phosphorylation of SnRK2s induced by 0.6 M mannitol or 50 μM ABA was detected with the anti-phospho-S175-SnRK2s antibody (top), using total proteins extracted from WT Col-0 and *bik1* mutant seedlings. Actin was used as the loading control (bottom). Quantification of pS175-SnRK2s (E) represents the ratio of band intensity of phosphorylated SnRK2s to that of actin. Data represent mean ± SD ($n = 3$ independent biological repeats). Different letters denote statistically significant differences according to one-way ANOVA followed by Tukey's test ($P < 0.05$). (G, H) Plant growth of Col-0 and *bik1* mutant seedlings, 12 days after the seedlings were transferred from ½ MS medium to ½ MS medium with or without 5 μM or 20 μM ABA (G). The rosette widths were quantified (H). Values are means ± SD ($n \geq 10$ seedlings). Two-way ANOVA followed by Tukey's test.

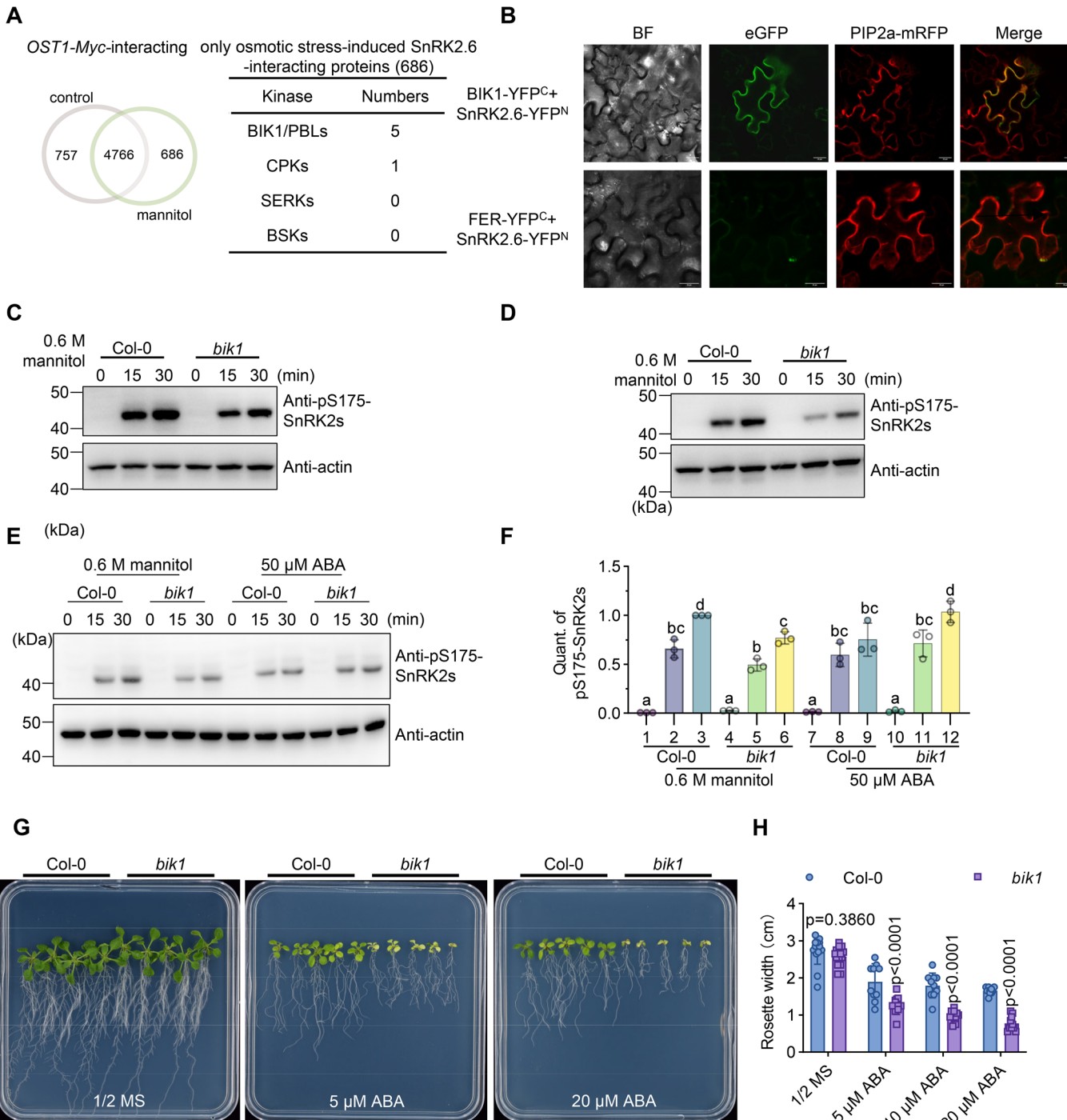

**A**

*OST1-Myc*-interacting

control 757 | 4766 | 686 mannitol

only osmotic stress-induced SnRK2.6-interacting proteins (686)

| Kinase | Numbers |
|--------|---------|
| BIK1/PBLs | 5 |
| CPKs | 1 |
| SERKs | 0 |
| BSKs | 0 |

**B**

BF | eGFP | PIP2a-mRFP | Merge

BIK1-YFP^C+ SnRK2.6-YFP^N

FER-YFP^C+ SnRK2.6-YFP^N

**C**

0.6 M mannitol

Col-0 | bik1

0 15 30 | 0 15 30 (min)

Anti-pS175-SnRK2s

Anti-actin

**D**

0.6 M mannitol

Col-0 | bik1

0 15 30 | 0 15 30 (min)

Anti-pS175-SnRK2s

Anti-actin

(kDa)

**E**

0.6 M mannitol | 50 µM ABA

Col-0 | bik1 | Col-0 | bik1

0 15 30 | 0 15 30 | 0 15 30 | 0 15 30 (min)

Anti-pS175-SnRK2s

Anti-actin

**F**

Quant. of pS175-SnRK2s

Col-0 | bik1 | Col-0 | bik1

0.6 M mannitol | 50 µM ABA

**G**

Col-0 | bik1 | Col-0 | bik1 | Col-0 | bik1

1/2 MS | 5 µM ABA | 20 µM ABA

**H**

● Col-0   ■ bik1

Rosette width (cm)

p=0.3860 | p<0.0001 | p<0.0001 | p<0.0001

1/2 MS | 5 µM ABA | 10 µM ABA | 20 µM ABA

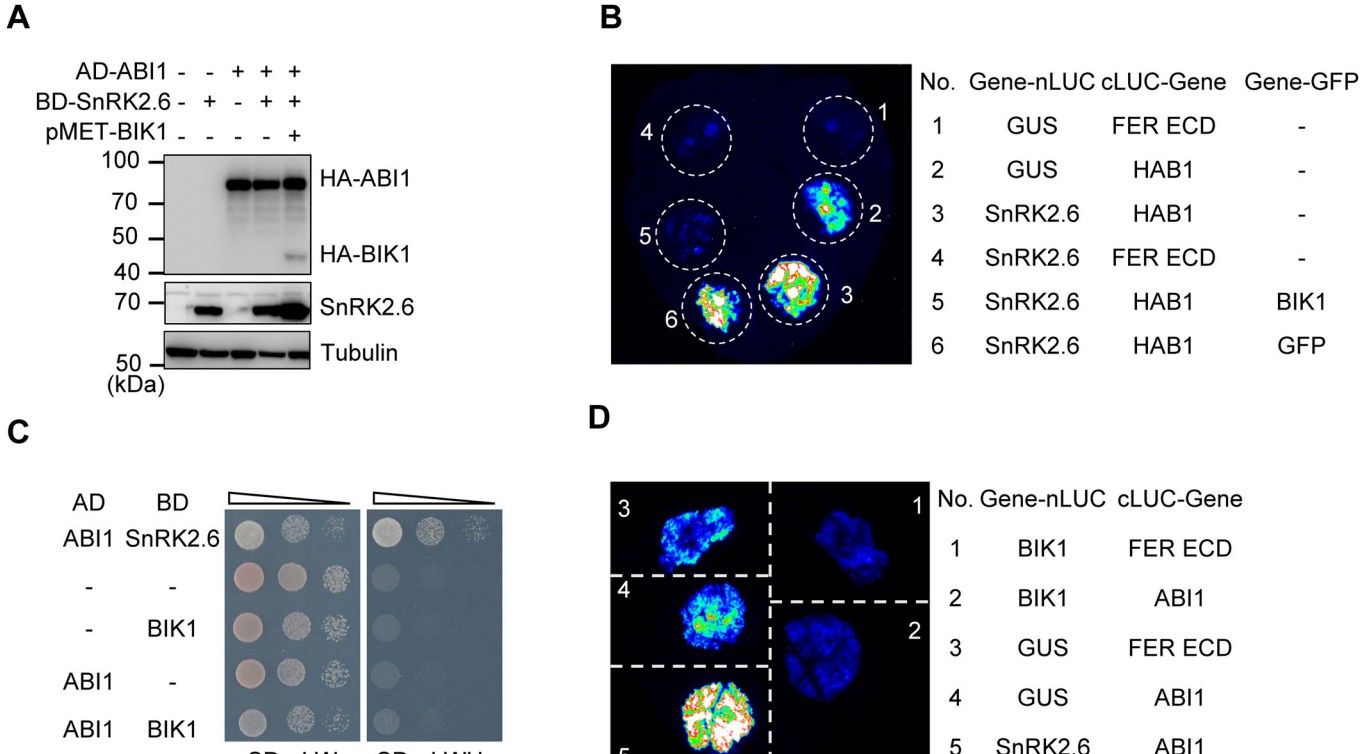

**Figure EV2.  BIK1 cannot interact with ABI1.**

(**A**) Protein expression levels of BD-SnRK2.6, AD-HA-ABI1, and HA-BIK1 in yeast cells for Y3H assay. Total proteins were extracted from the yeast AH109 cells and detected by western blot using anti-HA and anti-GAL4 DNA Binding Domain polyclonal antibodies. The anti-tubulin antibody was used as a loading control. Yeast growth in the Y3H assay is shown in Fig. 2A. (**B**) BIK1 can release SnRK2.6 from HAB1 binding in split luciferase (LUC) complementation assays. SnRK2.6 and HAB1 were fused to the split N- or C-terminal fragments of LUC (SnRK2.6-nLUC and cLUC-HAB1). The SnRK2.6-nLUC and cLUC-HAB1 combination was used as a positive control, and combinations of SnRK2.6-nLUC/cLUC-FER ECD, cLUC-HAB1/GUS-nLUC, and cLUC-HAB1/SnRK2.6-nLUC/GFP were used as negative controls. (**C, D**) BIK1 cannot interact with ABI1 in yeast two-hybrid (Y2H) (**C**) and split LUC assays (**D**). For Y2H assays, transformed yeast cells were grown on the nonselective medium lacking Leu and Trp (SD/ − LW) and the selective medium lacking Leu, Trp, and His (SD/ − LWH) (**C**). The AD-ABI1 and BD-SnRK2.6 combination was used as a positive control. For split LUC complementation assays, BIK1 and ABI1 were fused to the split N- or C-terminal fragments of LUC (BIK1-nLUC and cLUC-ABI1). The SnRK2.6-nLUC and ABI1-cLUC combination was used as a positive control, and combinations of BIK1-nLUC/cLUC-FER ECD and cLUC-ABI1/GUS-nLUC were used as negative controls.

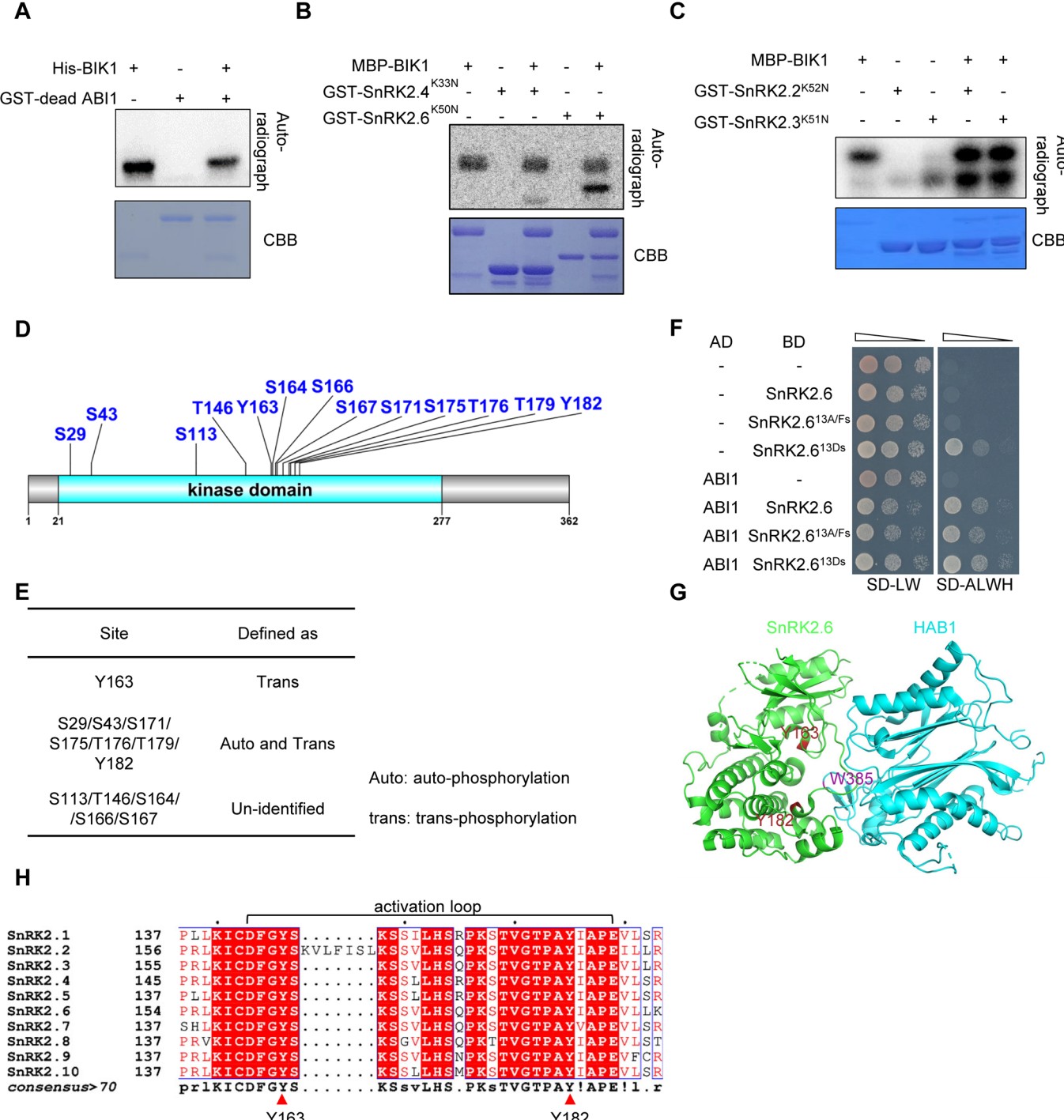

◀ **Figure EV3. BIK1 phosphorylates SnRK2.6 at multiple phosphosites.**

(A) Recombinant His-BIK1 cannot phosphorylate the catalytically inactive recombinant GST-ABI1 that was boiled for 10 min. Autoradiograph (top) and Coomassie blue staining (CBB, bottom) showed the phosphorylation and loading of proteins, respectively. (B, C) Phosphorylation of catalytically inactive variants of GST-tagged SnRK2.4 (K33N), SnRK2.6 (K50N), SnRK2.2 (K52N), and SnRK2.3 (K51N) by recombinant MBP–BIK1 during in vitro phosphorylation assays. Autoradiograph (top) and CBB staining (bottom) exhibited the phosphorylation and loading of purified recombinant proteins. (D) BIK1 phosphorylates SnRK2.6 at 13 putative phosphosites. These phosphosites were identified by mass spectrometry and distributed within the kinase domain, especially within the activation loop. (E) Summary of putative phosphosites in SnRK2.6 mediated by BIK1. (F) Interactions between ABI1 and wild-type and variant SnRK2.6 proteins, including the phospho-mimic (S/T/Y-to-D) mutations of all 13 putative BIK1-mediated phosphosites (SnRK2.6[13Ds]) and the non-phosphorylatable (S/T/Y-to-A/F) mutations (SnRK2.6[13A/Fs]). The transformed yeast cells were grown on the nonselective SD/ − LW medium and the selective SD/ − LWH medium. The AD-ABI1 and BD-SnRK2.6 combination was used as a positive control. The BD-SnRK2.6[13Ds] has strong self-activation in the Y2H assay. (G) Cartoon presentation of the SnRK2.6-HAB1 (PDB: 3UJG) structures highlighting the key tyrosine residues (red, Y163 and Y182) in SnRK2.6 and the tryptophan lock of HAB1 (magenta, W385). SnRK2.6 is shown in green, and HAB1 is shown in cyan. (H) Sequence alignment of the ten *Arabidopsis* SnRK2s showing the two conserved tyrosine residues. Y163 and Y182 residues are marked with red triangles.

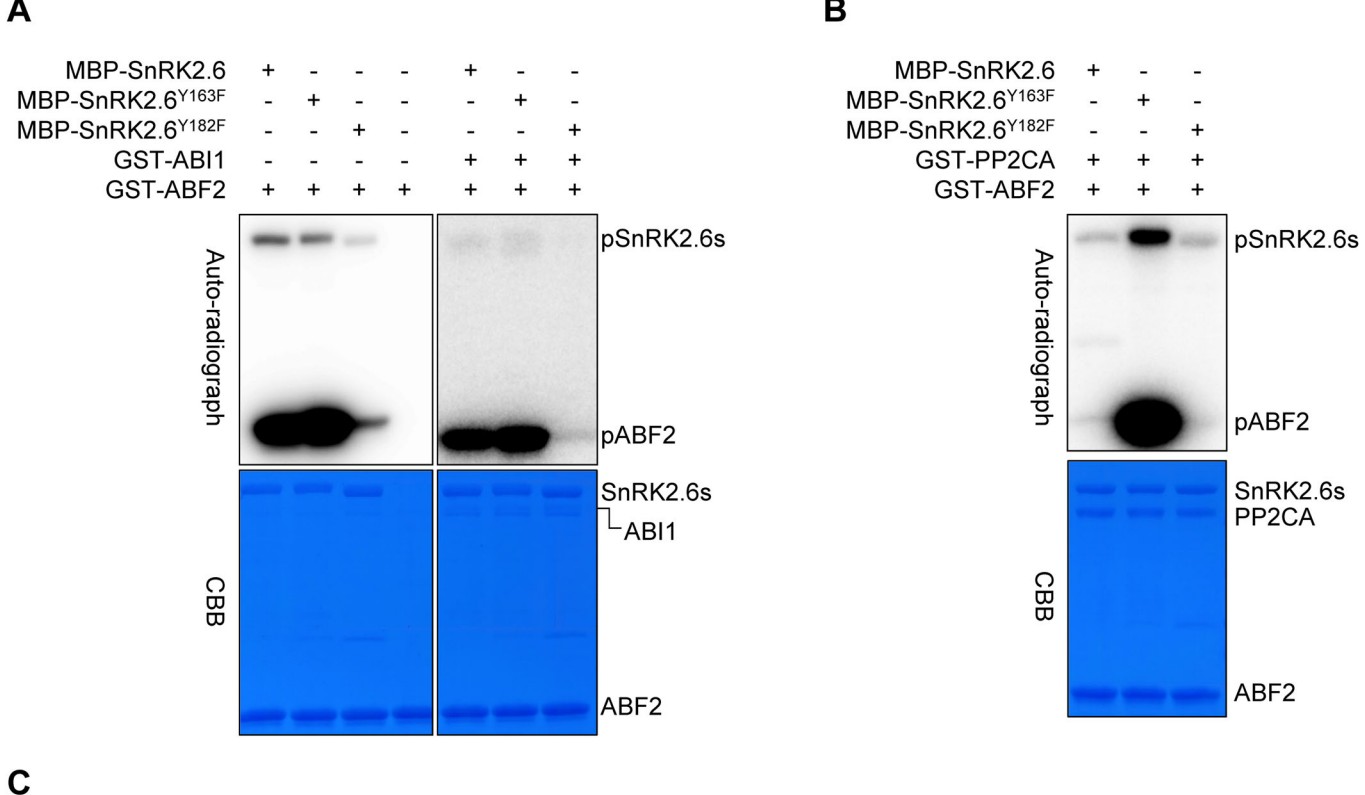

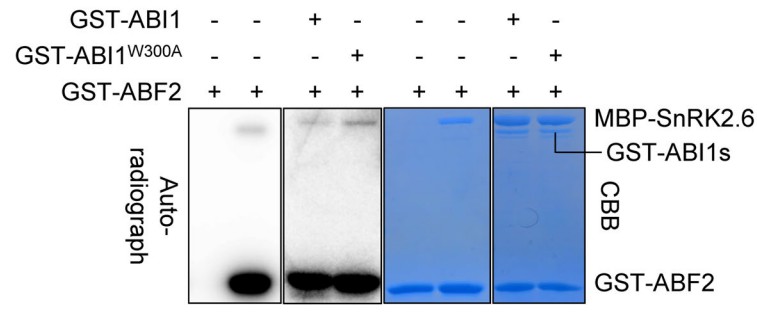

**Figure EV4. The Y163F mutation of SnRK2.6 reduces its inhibition by PP2Cs.**

(A, B) The Y163F mutation of SnRK2.6 reduces the inhibition of SnRK2.6 by ABI1 (A) and PP2CA (B) during in vitro phosphorylation assays. Autoradiography (top) and CBB staining (bottom) exhibited phosphorylation and loading of MBP-SnRK2.6, GST-ABI1, GST-PP2CA, and the GST-ABF2 fragment, respectively. The GST-ABF2 fragment was used as the substrate for SnRK2.6. (C) The tryptophan W300 of ABI1 is important for ABI1-mediated inhibition of SnRK2.6. The W300A mutation of ABI1 reduced its inhibition on SnRK2.6 during in vitro phosphorylation assays. The GST-ABF2 fragment was used as the substrate for SnRK2.6. Autoradiography (left) and CBB staining (right) exhibited phosphorylation and loading of proteins, respectively.

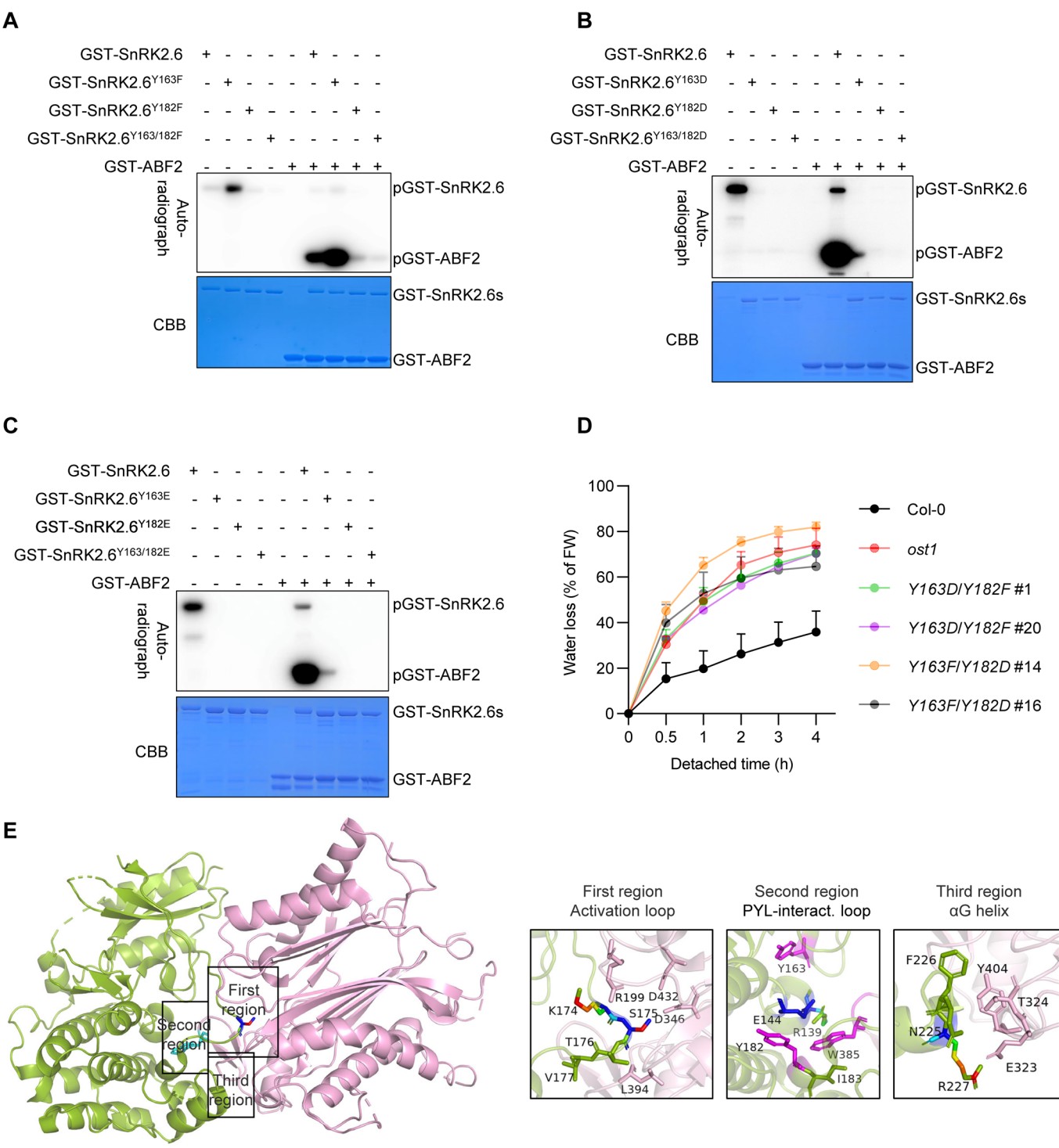

**Figure EV5.  The two tyrosine residues are critical for kinase activity and function of SnRK2.6.**

(A) Phosphorylation of the ABF2 fragment by wild-type and variant SnRK2.6 proteins, namely SnRK2.6$^{Y163F}$, SnRK2.6$^{Y182F}$, and SnRK2.6$^{Y163F/Y182F}$, during in vitro kinase assays. Autoradiography (top) and CBB staining (bottom) exhibited phosphorylation and loading of proteins, respectively. (B) Phosphorylation of the ABF2 fragment by SnRK2.6 and SnRK2.6$^{Y163D}$, SnRK2.6$^{Y182D}$, and SnRK2.6$^{Y163D/Y182D}$ during in vitro kinase assays. Autoradiography (top) and CBB staining (bottom) exhibited phosphorylation and loading of proteins, respectively. (C) Phosphorylation of the ABF2 fragment by SnRK2.6 and SnRK2.6$^{Y163E}$, SnRK2.6$^{Y182E}$, and SnRK2.6$^{Y163E/Y182E}$ during in vitro kinase assays. Autoradiography (top) and CBB staining (bottom) exhibited phosphorylation and loading of proteins, respectively. (D) Cumulative transpirational water loss from detached rosettes of five-week-old WT Col-0, *ost1-3* mutant, and *proSnRK2.6:SnRK2.6* transgenic plants in the *ost1-3* mutant background expressing mutated *SnRK2.6*, namely *SnRK2.6*$^{Y163D/Y182F}$, or *SnRK2.6*$^{Y163F/Y182D}$. Values are means ± SD ($n = 3$ independent experiments). (E) Structure (left) and zoomed images (right) exhibit the three major interfaces of the SnRK2.6-HAB1 complex (PDB: 3UJG). SnRK2.6 is shown in bean green, and HAB1 is shown in pink. Details of the SnRK2.6-HAB1 interfaces with key residues were shown with stick presentation (right).

