## [Peer Review File · The EMBO Journal]

Osmotic signaling releases PP2C-mediated inhibition of Arabidopsis SnRK2s via the receptor-like cytoplasmic kinase BIK1

Guo-Jun Li, Kong Chen, Shujing Sun, and Yang Zhao

Corresponding author(s): Yang Zhao (yangzhao@psc.ac.cn)

Review Timeline:

Submission Date:	8th May 24
Editorial Decision:	13th Jun 24
Revision Received:	9th Aug 24
Editorial Decision:	4th Sep 24
Revision Received:	10th Sep 24
Accepted:	30th Sep 24

Editor: William Teale

Transaction Report:

Dear Prof. Zhao,

Thank you again for the submission of your manuscript entitled "Osmotic signaling releases PP2C-mediated inhibition of SnRK2s via the receptor-like cytoplasmic kinase BIK1" (EMBOJ-2024-117817). We have now received the reports from two referees, which I copy below.

As you can see from their comments, both state that the study interesting and timely. However, both also raise questions (mainly over the way in which the mechanism you describe is integrated into the wider abiotic-signalling landscape) that will require your attention before your manuscript can be published in The EMBO Journal.

Based on the overall interest expressed in the reports, I would like to invite you to address the comments of all referees in a revised version of the manuscript. I should add that it is The EMBO Journal policy to allow only a single major round of revision and that it is therefore important to resolve the main concerns at this stage. I believe the concerns of the referees are reasonable and addressable, but please contact me if you have any questions, need further input on the referee comments or if you anticipate any problems in addressing any of their points. I am always available for a Zoom call to discuss the reports; please let me know if you would find this useful or not at this stage. Please also, follow the instructions below when preparing your manuscript for resubmission.

I would also like to point out that as a matter of policy, competing manuscripts published during this period will not be taken into consideration in our assessment of the novelty presented by your study ("scooping" protection). We have extended this 'scooping protection policy' beyond the usual 3 month revision timeline to cover the period required for a full revision to address the essential experimental issues. Please contact me if you see a paper with related content published elsewhere to discuss the appropriate course of action.

Again, please contact me at any time during revision if you need any help or have further questions.

Thank you very much again for the opportunity to consider your work for publication. I look forward to your revision.

Best regards,

William

William Teale, Ph.D.
Editor
The EMBO Journal

When submitting your revised manuscript, please carefully review the instructions below and include the following items:

- 1) a .docx formatted version of the manuscript text (including legends for main figures, EV figures and tables). Please make sure that the changes are highlighted to be clearly visible.
- 2) individual production quality figure files as .eps, .tif, .jpg (one file per figure).
- 3) a .docx formatted letter INCLUDING the reviewers' reports and your detailed point-by-point response to their comments. As part of the EMBO Press transparent editorial process, the point-by-point response is part of the Review Process File (RPF), which will be published alongside your paper.
- 4) a complete author checklist, which you can download from our author guidelines ([https://wol-prod-cdn.literatumonline.com/pb-assets/embo-site/Author Checklist%20-%20EMBO%20J-1561436015657.xlsx](https://wol-prod-cdn.literatumonline.com/pb-assets/embo-site/Author%20Checklist%20-%20EMBO%20J-1561436015657.xlsx)). Please insert information in the checklist that is also reflected in the manuscript. The completed author checklist will also be part of the RPF.
- 5) Please note that all corresponding authors are required to supply an ORCID ID for their name upon submission of a revised manuscript.
- 6) We require a 'Data Availability' section after the Materials and Methods. Before submitting your revision, primary datasets produced in this study need to be deposited in an appropriate public database, and the accession numbers and database listed

under 'Data Availability'. Please remember to provide a reviewer password if the datasets are not yet public (see <https://www.embopress.org/page/journal/14602075/authorguide#datadeposition>). If no data deposition in external databases is needed for this paper, please then state in this section: This study includes no data deposited in external repositories. Note that the Data Availability Section is restricted to new primary data that are part of this study.

Note - All links should resolve to a page where the data can be accessed.

8) For data quantification: please specify the name of the statistical test used to generate error bars and P values, the number (n) of independent experiments (specify technical or biological replicates) underlying each data point and the test used to calculate p-values in each figure legend. The figure legends should contain a basic description of n, P and the test applied. Graphs must include a description of the bars and the error bars (s.d., s.e.m.).

9) We would also encourage you to include the source data for figure panels that show essential data. Numerical data can be provided as individual .xls or .csv files (including a tab describing the data). For 'blots' or microscopy, uncropped images should be submitted (using a zip archive or a single pdf per main figure if multiple images need to be supplied for one panel). Additional information on source data and instruction on how to label the files are available at .

10) We replaced Supplementary Information with Expanded View (EV) Figures and Tables that are collapsible/expandable online (see examples in <https://www.embopress.org/doi/10.15252/embj.201695874>). A maximum of 5 EV Figures can be typeset. EV Figures should be cited as 'Figure EV1, Figure EV2" etc. in the text and their respective legends should be included in the main text after the legends of regular figures.

12) Our journal encourages inclusion of *data citations in the reference list* to directly cite datasets that were re-used and obtained from public databases. Data citations in the article text are distinct from normal bibliographical citations and should directly link to the database records from which the data can be accessed. In the main text, data citations are formatted as follows: "Data ref: Smith et al, 2001" or "Data ref: NCBI Sequence Read Archive PRJNA342805, 2017". In the Reference list, data citations must be labeled with "[DATASET]". A data reference must provide the database name, accession number/identifiers and a resolvable link to the landing page from which the data can be accessed at the end of the reference. Further instructions are available at .

We realize that it is difficult to revise to a specific deadline. In the interest of protecting the conceptual advance provided by the work, we recommend a revision within 3 months (11th Sep 2024). Please discuss the revision progress ahead of this time with the editor if you require more time to complete the revisions. Use the link below to submit your revision:

Referee #1:

Activation of SnRK2s under osmotic stress is critical for osmoregulation and plant drought resistance. The two-step activation of SnRK2s requires the release from PP2C-mediated inhibition and the following activation. This has been well-characterized in ABA signaling; however, the mechanism of SnRK2 release is poorly understood in osmotic stress signaling. Cell surface signaling has been linked to early osmotic stress signaling; however, how cell surface signaling regulates SnRK2 release and activation remains unclear. This manuscript characterized the function of BIK1-mediated tyrosine phosphorylation in releasing SnRK2.6 from PP2C-mediated inhibition under osmotic stress. The finding is exciting and some new findings are very interesting, including: 1) BIK1 releases SnRK2s via phosphorylation at two conserved tyrosines; 2) Release of SnRK2s from PP2Cs by BIK1 differs from that of PYLs; 3) The two tyrosines on SnRK2s are critical for ABI1-SnRK2 interaction and SnRK2 function; 4) The corresponding tryptophan lock of PP2Cs is essential for PP2C-SnRK2.6 binding; 5) BIK1 regulates plant osmotic stress responses. Generally, this manuscript uncovers an interesting mechanism for SnRK2 release under osmotic stress. I have some comments that improve the significance of the manuscript.

Major concerns:

1) The authors provide a new mechanism for SnRK2 release from PP2C inhibition under osmotic stress, which differs from the previously reported ABA receptor PYL-mediated SnRK2 release. They identified BIK1 from SnRK2-binding proteins using IP-MS analysis and found that lots of members in BIK1/PBL and CPK families interact with SnRK2.6 after mannitol treatment. They focused on the regulation of SnRK2s by BIK1 because of the reduced SnRK2 activation in *bik1* mutant seedlings under osmotic stress. The SnRK2 activity analysis is critical for this discovery and needs to be quantified.

Using multiple assays, they found that BIK1 is required and sufficient to release SnRK2s from PP2C binding and inhibition. However, whether other PBLs and CPKs participate in this process should be discussed.

2) Plants release and activate SnRK2 under osmotic stress via either an ABA-dependent or ABA-independent manner. Using the ABA-insensitive *abi1-1* mutation, the authors found that BIK1-mediated SnRK2 release differs from that of PYLs. My question is whether these two independent SnRK2 release mechanisms merge in plant osmoregulation. For example, whether ABA-mediated SnRK2 activation affected by the *bik1* mutation? Whether BIK1 mutation further dampen stress-responsive defects of ABA-related mutants? This should be evaluated and discussed since plants utilize multiple mechanisms for SnRK2 release and activation.

3) Other mechanisms for SnRK2 inhibition, release, and activation should be thoroughly discussed and compared with the BIK1-mediated SnRK2 release.

4) According to phenotype analyses of *bik1* mutant, BIK1 controls SnRK2 activation, ABA accumulation, stress-responsive gene expression, and seedling growth under osmotic stress. The authors mainly used mannitol treatment to mimic osmotic stress, and also used desiccation treatment in water loss analyses. To confirm these results, the authors should test whether *bik1* mutant seedlings also exhibit defects in response to other osmotic stress treatments such as sorbitol or NaCl. Treatment with osmotic stress gradients can improve the quality of the manuscript. The BIK1 activity analysis is also critical for this discovery and needs to be quantified.

5) The three major PP2C-SnRK2 interfaces described in the discussion part (lines 351-377) are somehow difficult to follow up. I suggest the authors add the structural modeling of the three interfaces to illustrate and visualize their findings.

Minor concerns:

- 1) Figure 4D. Can the authors simulate the effect of tyrosine phosphorylation on the interface between SnRK2.6 and HAB1?
- 2) Figure S1A, "OST1-Myc-interacting proteins", instead of "Super:OST1-Myc-interacting".
- 3) Figure S3F, the self-activation of SnRK2.6-13Ds in Y2H assay should be described.
- 4) Figure S3G, removes S113.
- 5) Figure S3I and S3J, what is the means for "Kn, Mp, Pp, Mp"? This should be stated in the legend.
- 6) The promoter length of OST1 used in this manuscript should be described in methods.

Referee #2:

This well-written manuscript by Li and colleagues resolved an important question in plant stress biology: how do plants release the PP2C-mediated SnRK2 inhibition before ABA accumulation under osmotic stress? In this manuscript, the receptor-like cytoplasmic kinase BIK1 is identified as an upstream regulator of osmotic stress signaling. This paper is focused on the release of SnRK2s. They found that BIK1 interacts with SnRK2.6 and releases SnRK2.6 from PP2C inhibition under osmotic stress, and this release differs from the PYL-mediated release of SnRK2s. They showed that BIK1 phosphorylates SnRK2.6 at two tyrosine residues, which may affect the docking of the tryptophan "lock" of PP2C into SnRK2.6. They demonstrated that the tyrosine phosphorylation of SnRK2.6 is critical for the kinase activity and its physiological function. This finding is exciting and answers the long-lasting question of osmotic stress signaling. I have the following suggestions to improve this discovery.

Major concerns:

- 1, This manuscript demonstrated that BIK1 releases SnRK2s from PP2C inhibition and this differs from the ABA- and PYL-mediated release of SnRK2s. In land plants, the ABA signaling functions together with the ancient osmotic signaling in mediating plant drought and osmotic stress responses. Therefore, it can be deduced that the ABA-independent release of SnRK2s (e.g., BIK1-mediated release) may function together with the PYL-mediated release of SnRK2s during plant osmotic stress responses. The sequential release of SnRK2s by BIK1 and ABA or PYLs should be discussed. It has been established that ABA signaling has a central role in plant growth, development, and stress responses. Does BIK1 function redundantly with ABA signaling in mediating plant growth and abiotic stress resistance? This may be resolved by crossing *bik1* with the ABA biosynthesis or signaling mutants.
- 2, Land plants evolve ABA receptors and signaling to mediate osmotic stress responses downstream of ABA (Sun et al., 2019). Osmotic stress shares signaling components with ABA in land plants, and are conserved in land plants and even in algae. Does BIK1 an evolutionarily conserved signaling component? This should be discussed. In figures S3I and S3J, the authors showed that SnRK2s and ABI1 are evolutionarily conserved regulators from algae to vascular plants. How about BIK1? Besides, figures S3I and S3J should be described and discussed, either in figure legends or the main text.
- 3, The conclusion of this work relies on the SnRK2-PP2C interactions in yeast and in plants, and SnRK2 activity in vivo and in vivo. In Figure 1E, the authors showed that SnRK2 activity is reduced in the *bik1* mutant seedlings under osmotic stress. This experiment is crucial but lacks replication or statistical analysis. I suggest the authors show the independent experiments as supplemental data. Otherwise, the experiments need to be quantified and statistically analyzed.
- 4, It is difficult to detect and to mimic tyrosine phosphorylation in plants. The authors tried to mimic tyrosine phosphorylation using Y-to-D and Y-to-E mutations. However, it seems that they failed to mimic tyrosine phosphorylation using these mutations. They proposed that the phosphorylation of the conserved tyrosine "could potentially increase the hydrophilicity and negative charges of the SnRK2.6". Please support this statement with modeling of structures.

Minor concerns:

- 1, Huang and colleagues reported that a plant NLR Sw-5b mimics PYLs in releasing SnRK2 from PP2C inhibition (Huang et al., 2024). It would be good to reference this paper and discuss the release of SnRK2s by PP2C-binding proteins besides PYLs.
- 2, line 149. Typos, "the interaction mainly occurs at the plasma member".
- 3, What is the mean for "short expo" and "long expo" in figure 4E and 4F? This should be provided in the figure legend.

4, Detailed information for SnRK2s and ABI1s in supplemental figures S3I and S3J should be provided in figure legends.

Huang, S., Wang, C., Ding, Z., Zhao, Y., Dai, J., Li, J., Huang, H., Wang, T., Zhu, M., Feng, M., Ji, Y., Zhang, Z., and Tao, X. (2024). A plant NLR receptor employs ABA central regulator PP2C-SnRK2 to activate antiviral immunity. *Nature Communications* 15, 3205.

Sun, Y., Harpazi, B., Wijerathna-Yapa, A., Merilo, E., de Vries, J., Michaeli, D., Gal, M., Cuming, A.C., Kollist, H., and Mosquna, A. (2019). A ligand-independent origin of abscisic acid perception. *Proc Natl Acad Sci USA* 116, 24892-24899.

Referee #1:

Activation of SnRK2s under osmotic stress is critical for osmoregulation and plant drought resistance. The two-step activation of SnRK2s requires the release from PP2C-mediated inhibition and the following activation. This has been well-characterized in ABA signaling; however, the mechanism of SnRK2 release is poorly understood in osmotic stress signaling. Cell surface signaling has been linked to early osmotic stress signaling; however, how cell surface signaling regulates SnRK2 release and activation remains unclear. This manuscript characterized the function of BIK1-mediated tyrosine phosphorylation in releasing SnRK2.6 from PP2C-mediated inhibition under osmotic stress. The finding is exciting and some new findings are very interesting, including: 1) BIK1 releases SnRK2s via phosphorylation at two conserved tyrosines; 2) Release of SnRK2s from PP2Cs by BIK1 differs from that of PYLs; 3) The two tyrosines on SnRK2s are critical for ABI1-SnRK2 interaction and SnRK2 function; 4) The corresponding tryptophan lock of PP2Cs is essential for PP2C-SnRK2.6 binding; 5) BIK1 regulates plant osmotic stress responses. Generally, this manuscript uncovers an interesting mechanism for SnRK2 release under osmotic stress. I have some comments that improve the significance of the manuscript.

Major concerns:

QUERY #1:

1) The authors provide a new mechanism for SnRK2 release from PP2C inhibition under osmotic stress, which differs from the previously reported ABA receptor PYL-mediated SnRK2 release. They identified BIK1 from SnRK2-binding proteins using IP-MS analysis and found that lots of members in BIK1/PBL and CPK families interact with SnRK2.6 after mannitol treatment. They focused on the regulation of SnRK2s by BIK1 because of the reduced SnRK2 activation in *bik1* mutant seedlings under osmotic stress. The SnRK2 activity analysis is critical for this discovery and needs to be quantified.

Using multiple assays, they found that BIK1 is required and sufficient to release SnRK2s from PP2C binding and inhibition. However, whether other PBLs and CPKs participate in this process should be discussed.

Response: We thank the reviewer for these suggestions. In the revised manuscript, we have included replications of experiments showing reduced SnRK2 activation in the *bik1* mutant under osmotic stress (Fig EV1C and D), and quantified SnRK2 activity in WT and the *bik1* mutant seedlings under osmotic stress (Fig 1E). We have also discussed the possibility of PBLs and CPKs in SnRK2 release and activation.

QUERY #2:

2) Plants release and activate SnRK2 under osmotic stress via either an ABA-dependent or ABA-independent manner. Using the ABA-insensitive *abil-1* mutation, the authors found that BIK1-mediated SnRK2 release differs from that of PYLs. My question is whether these two independent SnRK2 release mechanisms merge in plant osmoregulation. For example, whether ABA-mediated SnRK2 activation affected by the *bik1* mutation? Whether BIK1 mutation further dampen stress-responsive defects of ABA-related mutants? This should be evaluated and discussed since plants utilize multiple mechanisms for SnRK2 release and activation.

Response: We thank the reviewer for these constructive suggestions. We have evaluated SnRK2 activation by ABA in *bik1* mutant as suggested (Fig EV1E and F). Besides, the rosette growth of the *bik1* mutant was also severely inhibited in the medium with ABA (Fig EV1G and H). These results suggested that the *bik1* mutant is not defective in ABA responses and BIK1 is not required for ABA-induced SnRK2s activation.

ABA signaling is critical for plant growth and stress responses (Zhao *et al*, 2018). To evaluate whether BIK1 functions redundantly with ABA in mediating SnRK2 release and plant growth (Fig 3B), we generated the *bik1 aba2-1* double mutant and found that it has severe growth defects in soil (Fig 7H). However, we cannot get enough seeds to evaluate SnRK2 activation and other stress responses in the *bik1 aba2-1* double mutant.

QUERY #3:

3) Other mechanisms for SnRK2 inhibition, release, and activation should be thoroughly discussed and compared with the BIK1-mediated SnRK2 release.

Response: We thank the reviewer for this constructive suggestion and discussed other mechanisms for SnRK2 release and activation in the discussion section. The plant NLR immune receptor Sw-5b mimics PYLs via its nucleotide-binding (NB) domain and interacts with NbPP2C4 to release NbSnRK2s (Huang *et al*, 2024). In contrast, photoactivated phyA interacts with PP2Cs to repress PYL-mediated SnRK2 release (Li *et al*, 2023).

QUERY #4:

4) According to phenotype analyses of *bik1* mutant, BIK1 controls SnRK2 activation, ABA accumulation, stress-responsive gene expression, and seedling growth under osmotic stress. The authors mainly used mannitol treatment to mimic osmotic stress, and also used desiccation treatment in water loss analyses. To confirm these

results, the authors should test whether *bik1* mutant seedlings also exhibit defects in response to other osmotic stress treatments such as sorbitol or NaCl. Treatment with osmotic stress gradients can improve the quality of the manuscript. The BIK1 activity analysis is also critical for this discovery and needs to be quantified.

Response: We thank the reviewer for this suggestion and have performed new experiments as suggested. Four-day-old seedlings of WT and *bik1* mutant of similar sizes were transferred to plates containing ½ MS or ½ MS with mannitol, sorbitol, or NaCl (Figs 7A and B; Appendix Fig S4). The rosette growth of the *bik1* mutant was severely inhibited on the medium with mannitol (Figs 7A and B; Appendix Fig S4A and D), or with sorbitol and NaCl (Appendix Fig S4B-D).

We have also quantified BIK1 activity in *BIK1pro:BIK1-HA* transgenic seedling after osmotic stress treatment and found that BIK1 was strongly activated after 15 minutes of treatment with 600 mM mannitol (Fig 7F, lower panel). We further confirmed osmotic stress-induced BIK1 activation with *35S:BIK1-Myc* transgenic line, with *flg22* treatment as a positive control (Fig 7G).

QUERY #5:

5) The three major PP2C-SnRK2 interfaces described in the discussion part (lines 351-377) are somehow difficult to follow up. I suggest the authors add the structural modeling of the three interfaces to illustrate and visualize their findings.

Response: We thank the reviewer for this constructive suggestion. We have included the structural illustration of the three major interfaces of the SnRK2.6-HAB1 complex (Fig EV5E).

Minor concerns:

QUERY #6:

1) Figure 4D. Can the authors simulate the effect of tyrosine phosphorylation on the interface between SnRK2.6 and HAB1?

Response: We thank the reviewer for this constructive suggestion and have used AlphaFold 3 to predict the effect of tyrosine phosphorylation on the SnRK2.6-ABI1 interface. Structural simulation suggests that the phosphorylation of the conserved tyrosine increases the hydrophilicity and negative charges of the SnRK2.6

catalytic cleft and repels the docking of the nonpolar tryptophan lock of ABI1 (Fig 4G). However, the Y-to-D and Y-to-E point mutations of SnRK2.6 can hardly mimic the phosphorylation of the two conserved Tyr residues to repel the docking of the nonpolar tryptophan lock of ABI1 (Appendix Fig S3).

QUERY #7:

2) Figure S1A, "OST1-Myc-interacting proteins", instead of "Super:OST1-Myc-interacting".

Response: We have revised it as suggested.

QUERY #8:

3) Figure S3F, the self-activation of SnRK2.6-13Ds in Y2H assay should be described.

Response: We have revised it as suggested. The BD-SnRK2.6^{13Ds} fusion showed strong self-activation in the Y2H assay and did not exhibit improved growth in the presence of AD-ABI1 (Fig EV3F).

QUERY #9:

4) Figure S3G, removes S113.

Response: We have revised the figure as suggested (Fig EV3G).

QUERY #10:

5) Figure S3I and S3J, what is the means for "Kn, Mp, Pp, Mp"? This should be stated in the legend.

Response: We have revised the figure legend as suggested (Fig 3D and E).

QUERY #11:

6) The promoter length of OST1 used in this manuscript should be described in methods.

Response: We have revised the method as suggested. A 4,166-bp genomic DNA fragment containing 1461 bp upstream of the translation start site and 411 bp downstream of the stop codon of *SnRK2.6* was used.

Referee #2:

This well-written manuscript by Li and colleagues resolved an important question in plant stress biology: how do plants release the PP2C-mediated SnRK2 inhibition before ABA accumulation under osmotic stress? In this manuscript, the receptor-like cytoplasmic kinase BIK1 is identified as an upstream regulator of osmotic stress signaling. This paper is focused on the release of SnRK2s. They found that BIK1 interacts with SnRK2.6 and releases SnRK2.6 from PP2C inhibition under osmotic stress, and this release differs from the PYL-mediated release of SnRK2s. They showed that BIK1 phosphorylates SnRK2.6 at two tyrosine residues, which may affect the docking of the tryptophan "lock" of PP2C into SnRK2.6. They demonstrated that the tyrosine phosphorylation of SnRK2.6 is critical for the kinase activity and its physiological function. This finding is exciting and answers the long-lasting question of osmotic stress signaling. I have the following suggestions to improve this discovery.

Major concerns:

QUERY #1:

1, This manuscript demonstrated that BIK1 releases SnRK2s from PP2C inhibition and this differs from the ABA- and PYL-mediated release of SnRK2s. In land plants, the ABA signaling functions together with the ancient osmotic signaling in mediating plant drought and osmotic stress responses. Therefore, it can be deduced that the ABA-independent release of SnRK2s (e.g., BIK1-mediated release) may function together with the PYL-mediated release of SnRK2s during plant osmotic stress responses. The sequential release of SnRK2s by BIK1 and ABA or PYLs should be discussed. It has been established that ABA signaling has a central role in plant growth, development, and stress responses. Does BIK1 function redundantly with ABA signaling in mediating plant growth and abiotic stress resistance? This may be resolved by crossing *bik1* with the ABA biosynthesis or signaling mutants.

Response: We thank the reviewer for this constructive suggestion. We have discussed the correlation between ABA-independent and ABA-dependent release of SnRK2s as suggested. The ABA-independent release of SnRK2 (e.g., BIK1-mediated release) occurs earlier than ABA accumulation (Figs 1E, 7C and E) (Chen *et al.*, 2020), and may function redundantly with the PYL-mediated SnRK2 release in regulating plant growth and stress responses (Fig 7H and J).

ABA signaling is critical for plant growth and stress responses (Zhao *et al.*, 2018). To evaluate whether

BIK1 functions redundantly with ABA in mediating SnRK2 release and plant growth (Fig 3B), we generated the *bik1 aba2-1* double mutant and found that it has severe growth defects in soil (Fig 7H). However, we cannot get enough seeds to evaluate SnRK2 activation and other stress responses in the *bik1 aba2-1* double mutant.

QUERY #2:

2, Land plants evolve ABA receptors and signaling to mediate osmotic stress responses downstream of ABA (Sun et al., 2019). Osmotic stress shares signaling components with ABA in land plants, and are conserved in land plants and even in algae. Does BIK1 an evolutionarily conserved signaling component? This should be discussed. In figures S3I and S3J, the authors showed that SnRK2s and ABI1 are evolutionarily conserved regulators from algae to vascular plants. How about BIK1? Besides, figures S3I and S3J should be described and discussed, either in figure legends or the main text.

Response: We thank the reviewer for this suggestion and have discussed it as suggested. Besides SnRK2s and PP2Cs (Fig 4D and E), RLCK VIIs are also evolutionarily conserved kinases that originated in green algae (Ngou et al, 2024), and *MpPBLa* from *Marchantia polymorpha* can complement the *Arabidopsis bik1pbl1* mutant defect in chitin-induced ROS production (Chu et al, 2023). Future studies should investigate PBL function in SnRK2 release in green algae and early land plants.

There are three homologs of BIK1 in *Marchantia polymorpha*, including *MpPBLa*, *MpPBLb*, and *MpPBLc*, among which *MpPBLa* can also release SnRK2.6 from ABI1 inhibition in split-LUC assay in *N. benthamiana* (data not shown). Besides, BIK1 can release *MpSnRK2A* and *MpSnRK2B* from *MpABI1* interaction (data not shown). However, we have not got the triple mutant of *Mppbla/b/c* yet due to time limitations and technical challenges. Therefore, we discussed the putative role of BIK1 during evolution and will resolve this excellent question in the following study.

QUERY #3:

3, The conclusion of this work relies on the SnRK2-PP2C interactions in yeast and in plants, and SnRK2 activity in vitro and in vivo. In Figure 1E, the authors showed that SnRK2 activity is reduced in the *bik1* mutant seedlings under osmotic stress. This experiment is crucial but lacks replication or statistical analysis. I suggest the authors show the independent experiments as supplemental data. Otherwise, the experiments need to be quantified and statistically analyzed.

Response: We thank the reviewer for this suggestion. In the revised manuscript, we have included replications of experiments showing reduced SnRK2 activation in the *bik1* mutant under osmotic stress (Fig EV1C and D), and quantified SnRK2 activity in WT and the *bik1* mutant seedlings under osmotic stress (Fig 1E).

QUERY #4:

4, It is difficult to detect and to mimic tyrosine phosphorylation in plants. The authors tried to mimic tyrosine phosphorylation using Y-to-D and Y-to-E mutations. However, it seems that they failed to mimic tyrosine phosphorylation using these mutations. They proposed that the phosphorylation of the conserved tyrosine "could potentially increase the hydrophilicity and negative charges of the SnRK2.6". Please support this statement with modeling of structures.

Response: We thank the reviewer for this constructive suggestion and have used AlphaFold 3 to predict the effect of tyrosine phosphorylation on the SnRK2.6-ABI1 interface. Structural simulation suggests that the phosphorylation of the conserved tyrosine increases the hydrophilicity and negative charges of the SnRK2.6 catalytic cleft and repels the docking of the nonpolar tryptophan lock of ABI1 (Fig 4G). However, the Y-to-D and Y-to-E point mutations of SnRK2.6 can hardly mimic the phosphorylation of the two conserved Tyr residues to repel the docking of the nonpolar tryptophan lock of ABI1 (Appendix Fig S3).

Minor concerns:

QUERY #5:

1, Huang and colleagues reported that a plant NLR Sw-5b mimics PYLs in releasing SnRK2 from PP2C inhibition (Huang et al., 2024). It would be good to reference this paper and discuss the release of SnRK2s by PP2C-binding proteins besides PYLs.

Response: We thank the reviewer for this suggestion and have revised the manuscript as suggested.

QUERY #6:

2, line 149. Typos, "the interaction mainly occurs at the plasma member".

Response: We thank the reviewer for the careful reading and have revised the manuscript.

QUERY #7:

3, What is the mean for "short expo" and "long expo" in figure 4E and 4F? This should be provided in the figure legend.

Response: We have revised the figure legend (Fig 4H and I). "Short expo" means short exposure of the membrane, while "long expo" means long exposure.

QUERY #8:

4, Detailed information for SnRK2s and ABI1s in supplemental figures S3I and S3J should be provided in figure legends.

Response: We have revised the figure legend (Fig 4D and E).

References:

- Chen K, Gao J, Sun S, Zhang Z, Yu B, Li J, Xie C, Li G, Wang P, Song CP *et al* (2020) BONZAI proteins control global osmotic stress responses in plants. *Curr Biol* 30: 4815-4825 e4814
- Chu J, Monte I, DeFalco TA, Köster P, Derbyshire P, Menke FLH, Zipfel C (2023) Conservation of the PBL-RBOH immune module in land plants. *Curr Biol* 33: 1130-1137.e1135
- Huang S, Wang CL, Ding ZX, Zhao YQ, Dai J, Li J, Huang HN, Wang TK, Zhu M, Feng MF *et al* (2024) A plant NLR receptor employs ABA central regulator PP2C-SnRK2 to activate antiviral immunity. *Nat Commun* 15
- Li H, Zhou Y, Qin X, Peng J, Han R, Lv Y, Li C, Qi L, Qu G-P, Yang L *et al* (2023) Reconstitution of phytochrome A-mediated light modulation of the ABA signaling pathways in yeast. *Proc Natl Acad Sci U S A* 120: e2302901120
- Ngou BPM, Wyler M, Schmid MW, Kadota Y, Shirasu K (2024) Evolutionary trajectory of pattern recognition receptors in plants. *Nat Commun* 15
- Zhao Y, Zhang Z, Gao J, Wang P, Hu T, Wang Z, Hou YJ, Wan Y, Liu W, Xie S *et al* (2018) *Arabidopsis* duodecuple mutant of PYL ABA receptors reveals PYL repression of ABA-independent SnRK2 activity. *Cell Rep* 23: 3340-3351

Dear Yang,

We have now received re-review reports from two referees, which I have included below. As you will see, both support publication. Before I can finally accept the manuscript though, there are some remaining editorial points which need to be addressed. In this regard would you please:

- acknowledge grant number XDB27040107 and funder 'the Shanghai Center for Plant Stress Biology from the Chinese Academy of Sciences' in our online submission system,
- include up to five keywords,
- include a methods section in the main manuscript,
- rename the Conflict of Interests section as the 'Disclosure and competing interests statement',
- remove the author acknowledgement from the manuscript,
- include a callout in the manuscript text for dataset EV3,
- remove legends from "Supplemental methods" file and include them in a separate sheet in each Excel file,
- remove highlights from Appendix PDF,
- include a reagents and tools table,
- save source data files in a scheme of one figure/folder and then upload as .zip files. E.g. all the Source data files for figure 1 need to be saved in a single folder and this needs to be zipped and then uploaded as "SD figure 1.zip" file,
- include a data availability statement in the main manuscript file,
- define the annotated p values a/b/c/d/ab/bc as well as provide the exact p-values for the same in the legend of figure 1e, 7f; EV 1f; as appropriate,
- provide exact p values in the legends of figures 7b-e; EV 1h,
- correct the mismatch between the annotated p values in the figure legend and the annotated p values of figures 7b-f; EV 1h ,
- define box plots in terms of minima, maxima, centre, bounds of box and whiskers, and percentile in the legends of figure 7b,
- describe the nature of entity 'n' in the legend of figure EV 1f,
- define the measure of centre for the error bars in the legend of figures 7e, and
- use the following section order: title page with complete author information, abstract, keywords, introduction, results, discussion, methods, data availability section, acknowledgements, disclosure and competing interests statement, references, main figure legends, tables, expanded figure legends.

We include a synopsis of the paper (see <http://emboj.embopress.org/>). Please provide me with a general summary image, two-sentence summary statement and 3-5 bullet points that capture the key findings of the paper.

I am looking forward to receiving your revised manuscript.

EMBO Press is an editorially independent publishing platform for the development of EMBO scientific publications.

Best wishes,

William

William Teale, PhD
Editor
The EMBO Journal
w.teale@embojournal.org

- a point-by-point response to the referees' comments, with a detailed description of the changes made (as a word file).
- a word file of the manuscript text.
- individual production quality figure files (one file per figure)

- a complete author checklist, which you can download from our author guidelines (<https://www.embopress.org/page/journal/14602075/authorguide>).

- Expanded View files (replacing Supplementary Information)

We realize that it is difficult to revise to a specific deadline. In the interest of protecting the conceptual advance provided by the work, we recommend a revision within 3 months (3rd Dec 2024). Please discuss the revision progress ahead of this time with the editor if you require more time to complete the revisions. Use the link below to submit your revision:

Referee #1:

In this revised version of the paper, the authors have conducted a number of new experiments to prove their conclusion. I believe that the efforts of the authors and these new experiments can demonstrate the correctness of their conclusion. I do not have further comment.

Referee #2:

The authors answered all the questions that I raised first review.

QUERY #1:

- acknowledge grant number **XDB27040107** and funder 'the Shanghai Center for Plant Stress Biology from the Chinese Academy of Sciences' in our online submission system,

Response: We have acknowledged the grant number and the funder in the online submission system as suggested.

QUERY #2:

- include up to five keywords,
- include a methods section in the main manuscript,
- rename the Conflict of Interests section as the 'Disclosure and competing interests statement',
- remove the author acknowledgement from the manuscript,
- include a callout in the manuscript text for dataset EV3,
- remove legends from "Supplemental methods" file and include them in a separate sheet in each Excel file,
- remove highlights from Appendix PDF,

Response: We have revised the manuscript as suggested. We have removed the author contributions in the manuscript but reserved the acknowledgement.

QUERY #3:

- include a reagents and tools table,

Response: We have included the reagents and tools table.

QUERY #4:

- save source data files in a scheme of one figure/folder and then upload as .zip files. E.g. all the Source data files for figure 1 need to be saved in a single folder and this needs to be zipped and then uploaded as "SD figure 1.zip" file,

Response: We have revised the organization of source data as suggested.

QUERY #5:

- include a data availability statement in the main manuscript file,

Response: We have revised the manuscript as suggested.

QUERY #6:

- define the annotated p values a/b/c/d/ab/bc as well as provide the exact p-values for the same in the legend of figure 1e, 7f; EV 1f; as appropriate,
- provide exact p values in the legends of figures 7b-e; EV 1h,
- correct the mismatch between the annotated p values in the figure legend and the annotated p values of figures 7b-f; EV 1h ,

- define box plots in terms of minima, maxima, centre, bounds of box and whiskers, and percentile in the legends of figure 7b,
- describe the nature of entity 'n' in the legend of figure EV 1f,
- define the measure of centre for the error bars in the legend of figures 7e, and

Response: We have revised the figures and figure legends as suggested.

QUERY #7:

- use the following section order: title page with complete author information, abstract, keywords, introduction, results, discussion, methods, data availability section, acknowledgements, disclosure and competing interests statement, references, main figure legends, tables, expanded figure legends.

Response: We have organized the manuscript as suggested.

QUERY #8:

We include a synopsis of the paper (see <http://emboj.embopress.org/>). Please provide me with a general summary image, two-sentence summary statement and 3-5 bullet points that capture the key findings of the paper.

Response: We have prepared the general summary image.

We provided the following summary statement: “SnRK2 activation requires its release from PP2C inhibition, which is mediated by the BIK1-mediated tyrosine phosphorylation on SnRK2s under osmotic stress.”

We provided the following bullet points:

- 1) Activation of SnRK2s requires a two-step process: release of SnRK2s from PP2C inhibition and their subsequent activation.
- 2) BIK1 releases SnRK2.6 from PP2C binding and inhibition under osmotic stress.
- 3) Release of SnRK2s from PP2Cs by BIK1 differs from that of ABA and PYLs.
- 4) Two tyrosine residues phosphorylated by BIK1 are critical for SnRK2.6-ABI1 interaction.
- 5) The tryptophan lock of PP2C plays a critical role in PP2C-SnRK2.6 binding.

Dear Prof. Zhao,

I am pleased to inform you that your manuscript has been accepted for publication in the EMBO Journal.

Congratulations on the publication of this project!

Yours sincerely,

William Teale

William Teale, PhD
Editor
The EMBO Journal
w.teale@embojournal.org
